# Hydrogen peroxide release by bacteria suppresses inflammasome-dependent innate immunity

Saskia F. Erttmann [1] & Nelson O. Gekara [1,2]

Hydrogen peroxide ($H_2O_2$) has a major function in host-microbial interactions. Although most studies have focused on the endogenous $H_2O_2$ produced by immune cells to kill microbes, bacteria can also produce $H_2O_2$. How microbial $H_2O_2$ influences the dynamics of host-microbial interactions is unclear. Here we show that $H_2O_2$ released by *Streptococcus pneumoniae* inhibits inflammasomes, key components of the innate immune system, contributing to the pathogen colonization of the host. We also show that the oral commensal $H_2O_2$-producing bacteria *Streptococcus oralis* can block inflammasome activation. This study uncovers an unexpected role of $H_2O_2$ in immune suppression and demonstrates how, through this mechanism, bacteria might restrain the immune system to co-exist with the host.

[1] Laboratory for Molecular Infection Medicine Sweden (MIMS), Umeå Centre for Microbial Research (UCMR), Umeå University, 90 187 Umeå, Sweden. [2] Department of Molecular Biosciences, The Wenner-Gren Institute, Stockholm University, 106 91 Stockholm, Sweden. Correspondence and requests for materials should be addressed to N.O.G. (email: nelson.gekara@su.se)

Hydrogen peroxide ($H_2O_2$) is a ubiquitous metabolic byproduct of aerobic unicellular and multicellular organisms[1–4] that plays a major role in determining the outcome of host-microbial interactions[5]. Thus far, studies have focused on endogenous $H_2O_2$ produced by immune cells and its role in killing microbes or driving inflammatory processes[1,5–8]. However, $H_2O_2$ generation is not a preserve of eukaryotic host cells—many microbes do produce $H_2O_2$[1,4]. How $H_2O_2$ produced by microbes affect the dynamic of host-microbial interactions, and in particular the ability of immune cells to respond to microbes, remains unknown.

Accumulation of $H_2O_2$ to high levels can be toxic to both the host and the microbe. To mitigate these undesired effects, most organisms are equipped with enzymes such as catalase to neutralize $H_2O_2$[2]. However, some bacterial species lack catalase. One well-known example is *S. pneumoniae*. Consequently, a major hallmark of *S. pneumoniae* infection is massive production of $H_2O_2$, which can accumulate up to millimolars in in vitro cultures[9,10]. Considering that up to ~$10^8$ cfu/mL of *S. pneumoniae* has been documented in infected lungs[11], it is likely that the concentration of $H_2O_2$ in vivo, especially at the infection foci, is equally high hence can be detected in the breath of patients[12].

While $H_2O_2$ production is responsible for autolysis of bacteria at stationary phase, generally, *S. pneumoniae* is highly resistant to $H_2O_2$ via mechanisms not fully understood[13,14]. This resistance allows *S. pneumoniae* not only to outcompete other bacteria from the host niche but likely also to withstand $H_2O_2$ produced by immune cells. But how $H_2O_2$ produced by bacteria afffects the anti-microbial response of immune cells is unresolved.

Microbes—especially those that have established a long-standing co-existence with their hosts have evolved diverse mechanisms to mitigate anti-microbial innate immune defenses[13–15]. *S. pneumoniae* is a habitant of the upper respiratory tract and the most common causative agent of community-acquired pneumonia, a leading cause of death worldwide[16,17]. Infections with *S. pneumoniae* possibly also promote invasion of the airways by other pathogens[16,17].

How *S. pneumoniae* overcomes the innate immune system to persist in the host has remained unclear. Here we reveal that *S. pneumoniae* actively inhibits inflammasomes—key components of the innate immune system. This contributes to its ability to colonize the host but also renders immune cells unresponsive to other inflammasome stimuli during co-infections. We demonstrate that this phenomenon is due to *S. pneumoniae*-generated $H_2O_2$, which causes oxidative inactivation of inflammasomes. Further, we show that other $H_2O_2$-producing bacteria such as *Streptococcus oralis*, an oral commensal, similarly block inflammasomes. This study uncovers an unanticipated role of microbial $H_2O_2$ in innate immune suppression, and how this promotes host colonization by microbes.

## Results

### *S. pneumoniae* dampens inflammasome-dependent immunity via SpxB.
The major source of $H_2O_2$ generated by *S. pneumoniae* is via its pyruvate oxidase (SpxB)[9,18–20], which catalyzes the phosphate-dependent oxidation of pyruvate to form acetyl phosphate, $CO_2$ and $H_2O_2$ (Fig. 1a). SpxB has diverse and counter-intuitive effects on the host and the bacteria. While responsible for spontaneous bacterial lysis[19,21] and promotion of phagocytic uptake[22], SpxB confers survival fitness to S. pneumoniae in vivo[21] and contributes to resistance to oxidative stress[9]. Based on these divergent context-dependent effects of SpxB, we sought to investigate the impact of bacteria-derived $H_2O_2$ on the interaction between *S. pneumoniae* and a murine host.

Whereas the SpxB-deficient *S. pneumoniae* strain D39 (*S.p.* Δ*spxB*) that is highly defective in $H_2O_2$ production (Fig. 1b) exhibits a growth advantage over the wild-type strain in vitro (Fig. 1c), we find that upon intranasal inoculation in mice, *S.p.* Δ*spxB* causes severer clinical symptoms and is cleared faster without causing mortality (Fig. 1d–f). The severer symptoms and faster clearance of *S.p.* Δ*spxB* in vivo prompted us to investigate whether this mutant elicits a stronger anti-bacterial immune response. Indeed, although killed faster than wild-type *S. pneumoniae* D39 (*S.p.* WT) in vivo, *S.p.* Δ*spxB* elicits significantly higher levels of the inflammasome-dependent cytokine IL-1β (Fig. 1g). This is in contrast to the inflammasome-independent cytokine TNF-α, which, although showing a similar trend, is not significantly different between mice infected with wild-type or mutant bacteria (Fig. 1h). These results indicate that in the absence of SpxB, *S. pneumoniae* activates a stronger anti-bacterial immunity and in particular an inflammasome-dependent innate immune response.

### *S. pneumoniae* blocks inflammasomes via SpxB-generated $H_2O_2$.
Inflammasomes are intracellular multiprotein complexes that control the activation of Caspase-1, mediating pyroptotic cell death and the maturation of IL-1 family cytokines that are essential for optimal defence against pathogens including *S. pneumoniae*[23–26]. Generally, inflammasome activation involves two checkpoints: a priming step to induce the synthesis of pro-IL-1β and certain inflammasome components and a second step triggering the assembly and activation of inflammasome complexes[24]. To evaluate how *S. pneumoniae* modulates the second step of inflammasome activation, bone marrow-derived macrophages (BMDMs) were first primed with the TLR ligand lipopolysaccharide (LPS) and then infected with *S. pneumoniae* D39 for different durations. Thereafter, the magnitude and kinetics of Caspase-1 and IL-1β processing was compared with that by classical inflammasome activators. Here we show that although BMDMs infected with *S. pneumoniae* exhibit large amounts of pro-IL-1β and pro-Caspase-1, accumulation of the processed forms (IL-1β p17 and Casp-1 p20) is highly delayed and remains undetectable until 6 to 12 h post infection. Even at these delayed time points, the ratio of processed IL-1β and Caspase-1 to their precursors is very low (Supplementary Fig. 1a-c). This is in clear contrast to stimulation with ATP or nigericin (agonists of the NLRP3 inflammasome), *Salmonella* Typhimurium or *Pseudomonas aeruginosa* (activators of the NLRC4 inflammasome)[27,28], which elicit robust processing of Caspase-1 and IL-1β within 30 to 60 min (Supplementary Fig. 1d–g). This demonstrates that *S. pneumoniae* does not directly activate inflammasomes at the early stage of infection and that the weak response detected at later time points is likely a secondary outcome of danger signals such as ATP from dead or stressed cells or by pneumolysin extruded from lysed bacteria[29–32].

In view of these observations, we wondered whether such limited responses are due to inhibition of inflammasomes by *S. pneumoniae*. To directly test this idea, we pre-treated BMDMs with *S.p.* WT before challenging them with agonists for different inflammasomes including ATP, nigericin or *P. aeruginosa* (schematically illustration in Fig. 2a). *S.p.* WT dose-dependently inhibits the processing and secretion of Caspase-1, IL-1β and IL-18 (Fig. 2b–j). In contrast, TNFα is unaltered (Fig. 2k–m).

As *S.p.* Δ*spxB* induces elevated IL-1β levels in vivo (Fig. 1g), we tested whether *S. pneumoniae* blocks inflammasomes via SpxB. Indeed, inflammasome activation by ATP and *P. aeruginosa* is highly inhibited in the presence of *S.p.* WT (Fig. 3a–c) but not *S.p.* Δ*spxB*, except upon complementation with a plasmid carrying the *spxB* gene (Fig. 3d, e).

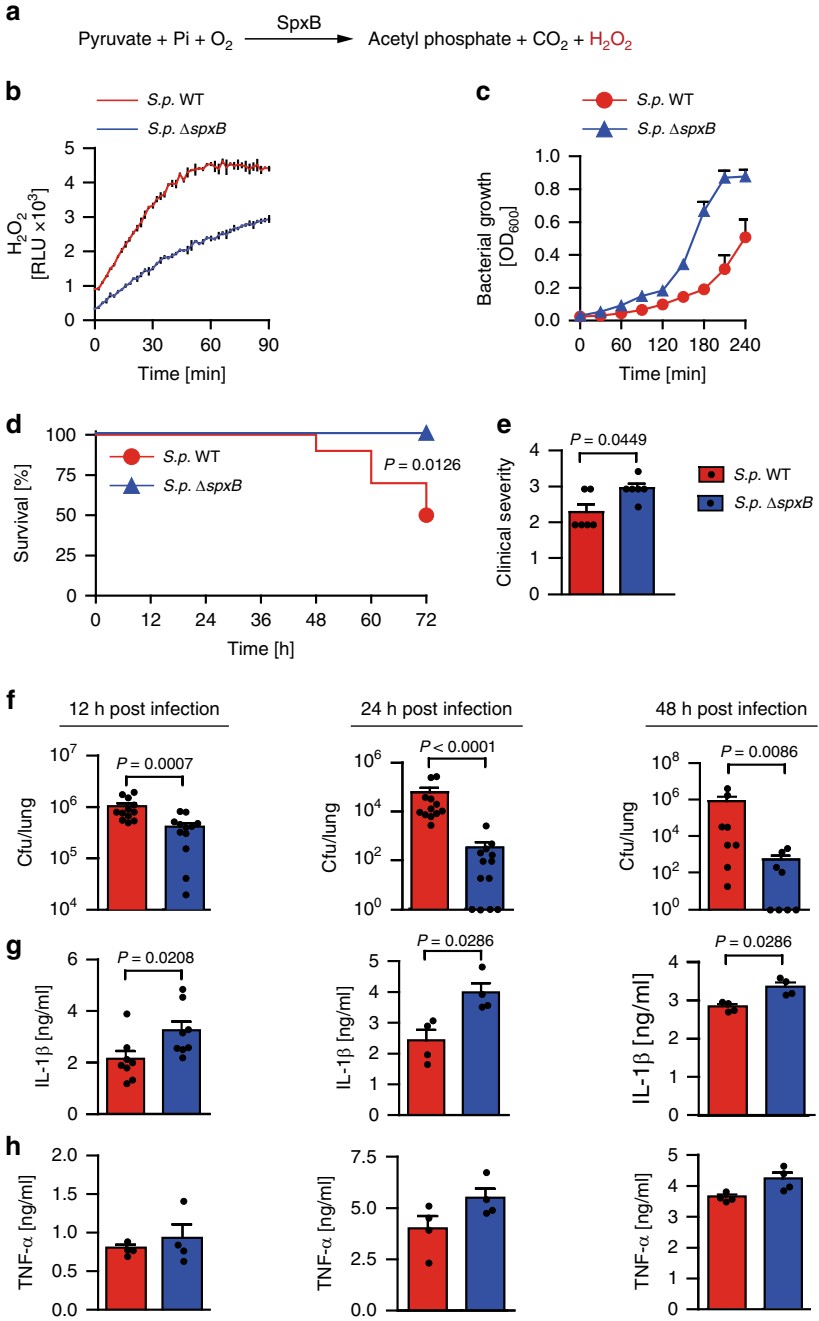

**Fig. 1** Pyruvate oxidase SpxB dampens inflammasome-dependent cytokine response promoting *S. pneumoniae* survival. **a** The pyruvate oxidase (SpxB)-mediated $H_2O_2$ generation reaction in *S. pneumoniae*. **b** $H_2O_2$ release by wild-type *S. pneumoniae* D39 (*S.p.* WT) and *S.p.*Δ*spxB*. **c** In vitro growth rate of *S.p.* WT (D39) and *S.p.* Δ*spxB*. **b** and **c** are representative of 3 independent experiments performed in triplicates; data shown as mean ± standard deviation (±s.d.). **d** Survival of WT mice after intranasal infection ($1–2 \times 10^7$ cfu/mouse) with *S.p.* WT (D39) or *S.p.* Δ*spxB* analysed by the Kaplan–Meier method, with $n = 10$ animals per group. *P* value determined by Gehan-Breslow-Wilcoxon test. **e** Clinical severity of WT mice infected with *S.p.* WT (D39) or *S.p.* Δ*spxB* ($1–2 \times 10^7$ cfu/mouse) for 6 h ($n = 6$ animals per group). *P* value determined by Mann Whitney test. **f** *S.p.* WT (D39) and *S.p.* Δ*spxB* counts (cfu) in the lungs of WT mice 12, 24 and 48 h after intranasal infection ($1–2 \times 10^7$ cfu/mouse). **g** IL-1β and **h** TNF-α in the lung fluid of WT mice 12, 24 and 48 h after intranasal infection ($1–2 \times 10^7$ cfu/mouse) with *S.p.* WT (D39) or *S.p.* Δ*spxB*. Results in **f** to **h** are from 2–3 independent experiments with a total of 4–12 animals per group; data shown as the mean ± standard error of the mean (±s.e.m.). *P* value determined by Mann Whitney test. Source data are provided as a Source Data file

However, pre-infection of cells with *S. pneumoniae* does not inhibit the induction of Pro-IL-1β or TNFα by the TLR ligand LPS, thus ruling out priming as the target of *S. pneumoniae*-mediated inflammasome inhibition (Supplementary Fig. 2a–f).

Further, LDH release and bacterial uptake assays show that the diminished inflammasome response is not due to loss of phagocytic activity or viability (Supplementary Fig. 2g–i). Notably, long-term infection of macrophages reveals that the SpxB

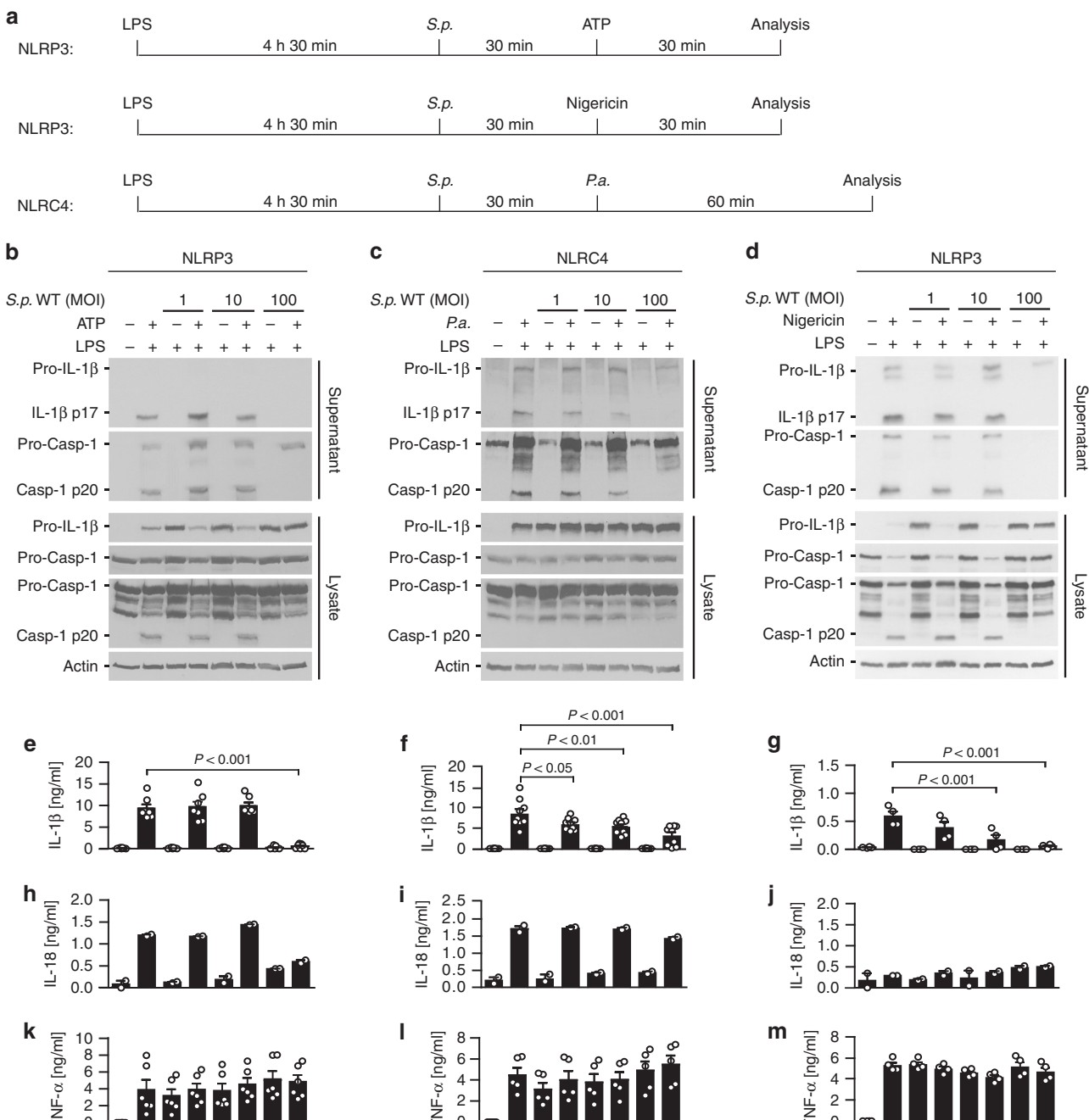

**Fig. 2** *S. pneumoniae* impairs NLRP3- and NLRC4-dependent inflammasome activation. **a** Schematic diagrams of experimental setups used in **b** to **m**. Immunoblots of Caspase-1 and IL-1β processing in LPS-primed BMDMs pre-treated with increasing *S.p.* WT (D39) doses (MOI 1, 10 or 100) for 30 min before stimulation with **b** ATP for 30 min, **c** *P. aeruginosa* (*P.a.*, MOI 20) for 60 min or **d** nigericin for 30 min. Results are representative of at least 4 independent experiments. ELISA analysis of corresponding supernatants for **e**–**g** IL-1β, **h**–**j** IL-18 and **k**–**m** TNF-α. Results are obtained from at least two independent experiments; data are shown as mean ± s.e.m. *P* values determined by one-way ANOVA followed by Bonferroni post-test. Source data are provided as a Source Data file

mutant bacteria have an intrinsically increased capacity to activate the inflammasome when compared to WT bacteria (Supplementary Fig. 3). Together, these data demonstrate that *S. pneumoniae* actively blocks inflammasome activation.

Next, we sought to determine the bacterial factors responsible for inflammasome activation by *S. pneumoniae*. Pneumolysin (PLY) is an intracellular thiol-dependent pore forming toxin released into the environment upon bacterial lysis[33,34]. In view of previous studies[29–32,35,36], we compared the ability of *S.p.* WT

and *ply*-deficient *S. pneumoniae* (*S.p.* Δ*ply*) to induce inflammasome activation. As observed before, short (1 h) infection with *S.p.* WT or *S.p.* Δ*ply* fail to induce inflammasome activation. However, long-term (12 h) infection lead to a modest response by *S.p.* WT but not *S.p.* Δ*ply*, confirming the importance of PLY in *S. pneumoniae*-mediated inflammasome activation (Supplementary Fig. 4a). As a proxy to evaluate inflammasome activation by PLY and how this is affects by *S. pneumoniae* we used a highly purified listeriolysin O (LLO)—a homologous pore-forming

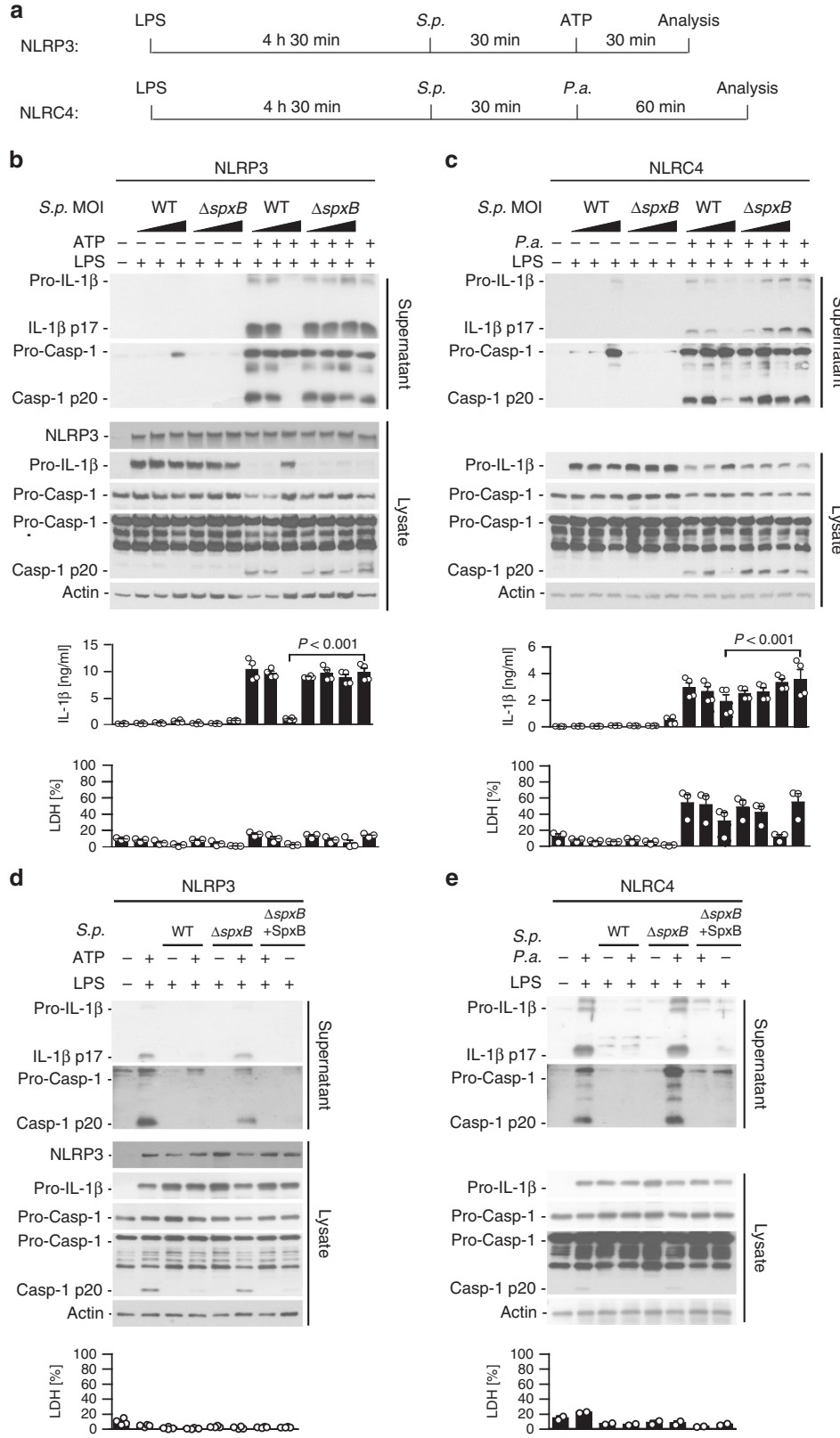

toxin. LLO induces robust inflammasome activation that is inhibited by *S.p.* WT but not *S.p.* Δ*spxB* (Supplementary Fig. 4b–d). Similar to PLY, LLO is a thiol-dependent toxin hence highly sensitive to oxidation[37]. Therefore, we employed catalase, an $H_2O_2$-neutralizing enzyme, to reduce the accumulation of reactive oxygen species. Treatment with catalase enhances

LLO-mediated inflammasome activation and increases the ability of *S.p.* WT to induce inflammasome activation (Supplementary Fig. 4e, f). Hence, we concluded that the restrained inflammasome response elicited by *S. pneumoniae* is likely due to oxidative inactivation of host signaling molecules as well as the bacterial inflammasome agonist PLY.

**Fig. 3** *S. pneumoniae* blocks multiple inflammasome pathways through its SpxB activity. **a** Schematic diagram of experimental setups used in (**b–e**). **b**, **c** Immunoblots of Caspase-1 and IL-1β in cell lysates and supernatants of LPS-primed BMDMs pre-treated with increasing MOIs (1, 10, 100) of *S.p.* WT (D39) or *S.p.* Δ*spxB* and then stimulated with agonists for NLRP3 (ATP, 5 mM, 30 min), or NLRC4 (*P.a.*, MOI 20, 60 min). ELISA for IL-1β secretion and lactate dehydrogenase (LDH) assay in corresponding supernatants. The results are representative of four independent experiments. *P* values determined by one-way ANOVA followed by Bonferroni post-test. **d**, **e** Caspase-1 and IL-1β processing and LDH release of LPS-primed BMDMs pre-treated with *S.p.* WT (D39), *S.p.* Δ*spxB* or *S.p.* SpxB-complemented Δ*spxB* (Δ*spxB* + SpxB) (MOI 50) for 30 min before **d** stimulation with 5 mM ATP for 30 min or **e** infection with *P.a.* (MOI 20) for 60 min. The results are representative of four or two independent experiments, respectively. LDH release is depicted as the mean ± s.e.m. Source data are provided as a Source Data file

**S. pneumoniae blocks the assembly of inflammasome complexes**. ASC oligomerization is a critical step in the assembly and activation of inflammasome complexes[38,39]. To test whether *S. pneumoniae* D39 affects this step, we employed differential centrifugation and chemical cross-linking[40] to isolate and analyze ASC oligomers. LPS-primed BMDMs pre-treated with *S.p.* WT exhibit severely impaired ASC oligomerization and diminished processing of Caspase-1 and IL-1β in response to ATP or *P. aeruginosa*. This is in contrast to *S.p.* Δ*spxB*-pre-treated cells that do not exhibit inhibited ASC oligomerization (Fig. 4a, b), except upon complementation with the plasmid carrying the *spxB* gene (Supplementary Fig. 5a, b). To verify these findings, BMDMs were fluorescently labelled with the Caspase-1 fluorescent activity-based probe FLICA (FAM-YVAD-FMK) and an anti-ASC antibody and then analyzed microscopically for ASC/Caspase-1 complexes (specks). BMDMs pre-treated with WT *S. pneumoniae* D39 exhibit highly reduced ASC/Caspase-1 speck formation in response to ATP or *P. aeruginosa* (Fig. 4c–f). To determine whether inflammasome inhibition is a general feature of *S. pneumoniae*, we tested other *S. pneumoniae* strains including A66.1 (serotype 3), TIGR4 (serotype 4), and the unencapsulated strain R6. All such strains inhibit ASC oligomerization and processing of Caspase-1 and IL-1β (Supplementary Fig. 6a–d), thereby demonstrating that the observed inflammasome inhibition is not strain-specific.

**High H₂O₂-producing bacteria inhibit inflammasome activation**. To further examine the generality of inflammasome inhibition by H₂O₂-producing bacteria, we tested other bacterial species including the oral commensals *Streptococcus oralis* and *Streptococcus sobrinus*, and the human and pig pathogens *Streptococcus pyogenes* and *Streptococcus suis*, respectively. By comparing the magnitude of H₂O₂ production, we can classify these bacteria into high, intermediate and low H₂O₂ producers (Fig. 5a). When tested on BMDMs, the high H₂O₂ producers *S. pneumoniae* and *S. oralis* strongly inhibit inflammasome activation. The intermediate H₂O₂ producers *S.p.* Δ*spxB*, *S. sobrinus* and *S. suis* neither inhibit nor induce inflammasome activation. In contrast, the low H₂O₂ producer *S. pyogenes* does not inhibit but rather induces inflammasomes (Fig. 5b–i). These data demonstrate that inflammasome inhibition by different bacterial species correlates with the strength of H₂O₂ release.

To verify whether the observed inflammasome inhibition is mediated by H₂O₂, we used catalase. Catalase promotes the in vitro growth of the high H₂O₂ producers WT *S. pneumoniae* and *S. oralis*, but not the intermediate or low H₂O₂ producers *S.p.* Δ*spxB*, *S. sobrinus*, *S. suis* or *S. pyogenes* (Supplementary Fig. 7a–c). Notably, catalase rescues inflammasome inhibition by *S.p.* WT (Fig. 6a, b and Supplementary Fig. 7d–o) and *S. oralis*, but has no effect on the IL-1β response by *S. sobrinus* or TNF-α production (Supplementary Fig. 7d–o). Moreover, *S. pneumoniae* is unable to block inflammasome activation in the presence of catalase in human PBMCs (Supplementary Fig. 8a–h). In contrast, inflammasome inhibition can be recapitulated by direct addition of H₂O₂ to BMDMs (Fig. 6c, d). Titration of the H₂O₂

amount reveals that 50–100 μM H₂O₂ are sufficient to inhibit Caspase -1 as well as Caspase -3 and Caspase -8 (Supplementary Fig 9a). In contrast to *S.p.* Δ*spxB*, *S.p.* WT and the *S.p.* Δ*spxB* carrying the SpxB complementing plasmid release similarly high amounts of H₂O₂ into the cell culture medium during infection (Supplementary Fig. 9b, c).

The above data unequivocally demonstrate that H₂O₂ is a potent inhibitor of inflammasomes. This led us to hypothesize that such inhibition of inflammasomes is due to oxidation of inflammasome components. Protein carbonylation is the most general marker of severe oxidative protein damage. It involves covalent adduction of carbonyl groups (e.g. aldehydes and ketones) on oxidised side chains of susceptible amino acid residues such as cysteine, proline, arginine, lysine and threonine[41]. Carbonylated proteins can be labelled with dinitrophenylhydrazine and detected using anti-dinitrophenol (DNP) antibodies. Immunoblotting of DNP-derivatized cell extracts reveals increased protein carbonylation in macrophages exposed to *S.p.* WT (Supplementary Fig. 10a, b), indicating that H₂O₂ production by *S. pneumoniae* causes oxidation of proteins in host cells. Given that oligomerization of ASC, a key inflammasome component, is impaired in the presence of *S. pneumoniae* or H₂O₂ (Fig. 4, Fig. 6, Supplementary Fig. 5 and Supplementary Fig 6), we next tested whether ASC is among those proteins oxidised by *S. pneumoniae*. Immunoprecipitation and immunoblot analyses of DNP-derivatized cell extracts reveal that ASC is carbonylated upon exposure to WT *S. pneumoniae* and that such oxidation is associated with defective Caspase-1 and IL-1β processing (Supplementary Fig. 10c). Based on the results, we conclude that H₂O₂-mediated inflammasome inhibition is due to the oxidative inactivation of ASC and most likely other components including the oxidation-sensitive cysteine-rich caspases[42,43].

**H₂O₂ blocks inflammasome-dependent bacterial clearance**. Considering that *S.p.* Δ*spxB* elicits a stronger inflammasome response and is cleared faster from mice (Fig. 1), we investigated whether inflammasome activation contributes to enhanced clearance of *S.p.* Δ*spxB* using *ASC*⁻/⁻ mice. At early time points, no ASC-dependent clearance of *S.p.* WT D39 is evident. However, as previously observed[25,26], *ASC*⁻/⁻ mice exhibit higher bacterial burden than wild-type mice 48 h after infection (Fig. 7a–c). ASC-dependent clearance of *S.p.* WT at later rather than early time points is consistent with its H₂O₂-mediated inhibitory effects on inflammasomes. Hence, the elicited inflammasome response is both low and delayed. In contrast to *S.p.* WT, ASC-dependent clearance of *S.p.* Δ*spxB* is observed as early as 12 and 24 h after infection (Fig. 7d–f). Noteworthy, although carrying a higher bacterial burden—consistent with the lack of inflammasome-dependent inflammation—the *ASC*⁻/⁻ mice exhibit lower clinical severity and body temperature changes (Fig. 7g, h).

In addition to metabolism and H₂O₂ production, SpxB also plays a role in other aspects of bacterial physiology[9,44,45], which together with the modulation of inflammasomes may contribute

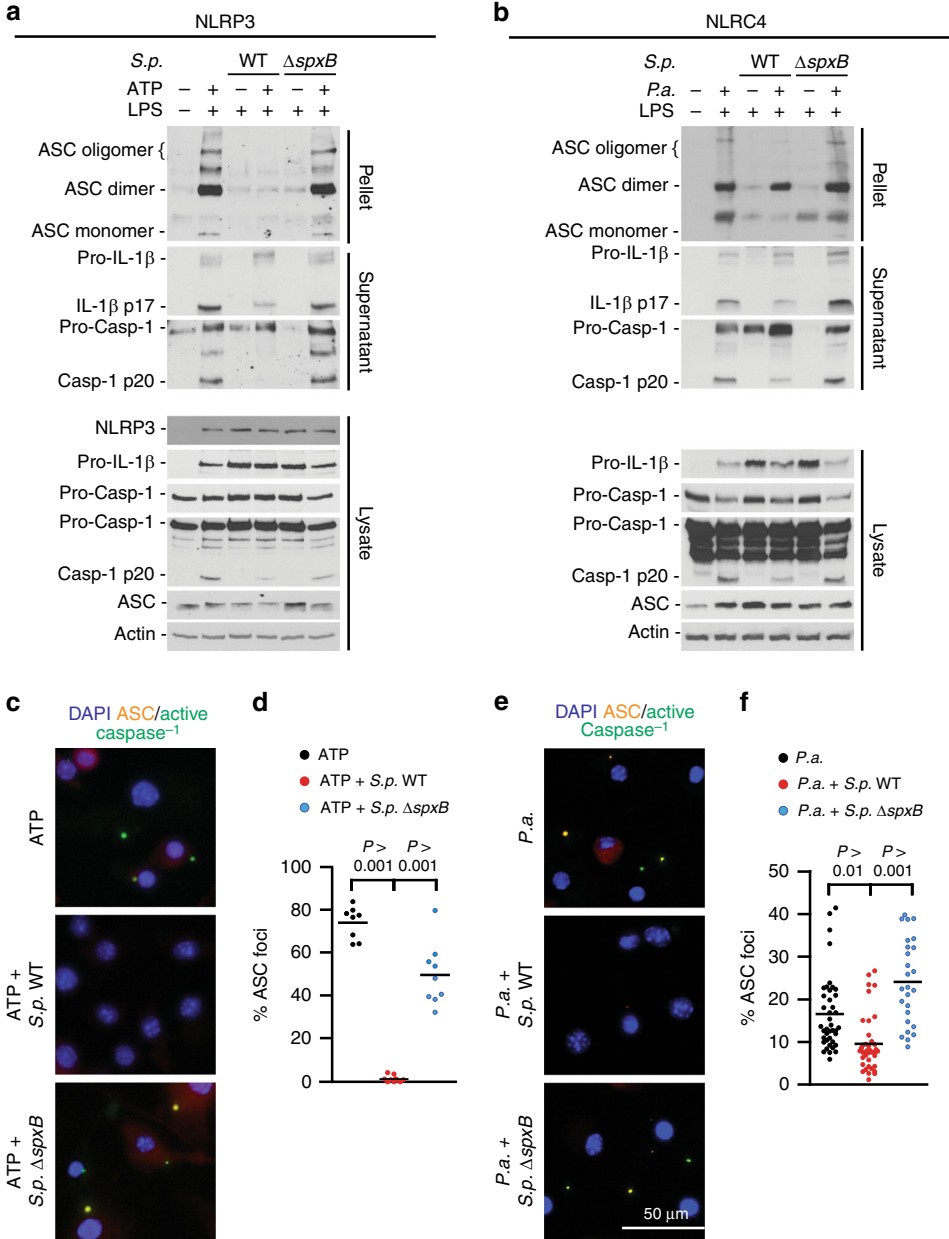

**Fig. 4** Inflammasome inhibition by *S. pneumoniae* involves defects in ASC complex formation. **a**, **b** Immunoblot analyses of ASC oligomers or Caspase-1 and IL-1β processing in LPS-primed BMDMs pre-treated with *S.p.* WT or *S.p.* Δ*spxB* (D39, MOI 50, 30 min) before stimulation with **a** ATP for 30 min (NLRP3) or **b** *P.a.* (NLRC4, MOI 20) for 60 min. Results are representative of 3 independent experiments. **c–f** Microscopic visualisation of ASC/active Caspase-1 complexes (red/green; overlay yellow) in LPS-primed BMDMs pre-treated with *S.p.* WT or *S.p.* Δ*spxB* (D39, MOI 50) and subsequently stimulated with **c**, **d** ATP (5 mM, 30 min) or **e**, **f** *P.a.* (MOI 20, 60 min). Panels **d** and **f** depict corresponding percentages of cells in **c** and **e**, respectively, containing ASC/active Caspase-1 specks (determined by enumerating at least 100 cells per sample). Scale bar: 50 μm. Results in **d** and **f** are from three independent experiments. The data are shown as the mean ± s.e.m. *P* values determined by one-way ANOVA followed by Bonferroni post-test. Source data are provided as a Source Data file

to the in vivo attenuation of *S.p.* Δ*spxB*. Therefore, to specifically evaluate the contribution of $H_2O_2$ on the infection outcome, we asked whether in vivo neutralization of $H_2O_2$ by catalase could similarly increase *S. pneumoniae*-induced inflammation and whether this in turn could promote bacterial clearance. In agreement, when intranasally co-inoculated with catalase, *S.p.* WT elicits severer clinical symptoms and is cleared faster (Fig. 7i–k), while clearance of *S.p.* Δ*spxB* is unaltered (Fig. 7l–n). This independently confirms that $H_2O_2$ release by *S. pneumoniae* does inhibit innate immune activation and that this contributes to host colonization as schematically summarized in Fig. 7o.

## Discussion

Here we report that in order to persist in the host *S. pneumoniae* interferes with a key arm of the innate immune response—inflammasome activation. We show that this feature is due to the ability of *S. pneumoniae* to release large quantities of hydrogen peroxide ($H_2O_2$) thereby causing oxidative inactivation of inflammasomes in immune cells. Further, we demonstrate that inflammasome inhibition is a common feature of other high $H_2O_2$-producing bacterial species. These findings are significant. Firstly, they uncover a new mechanism of innate immune subversion by bacteria. Secondly, they highlight

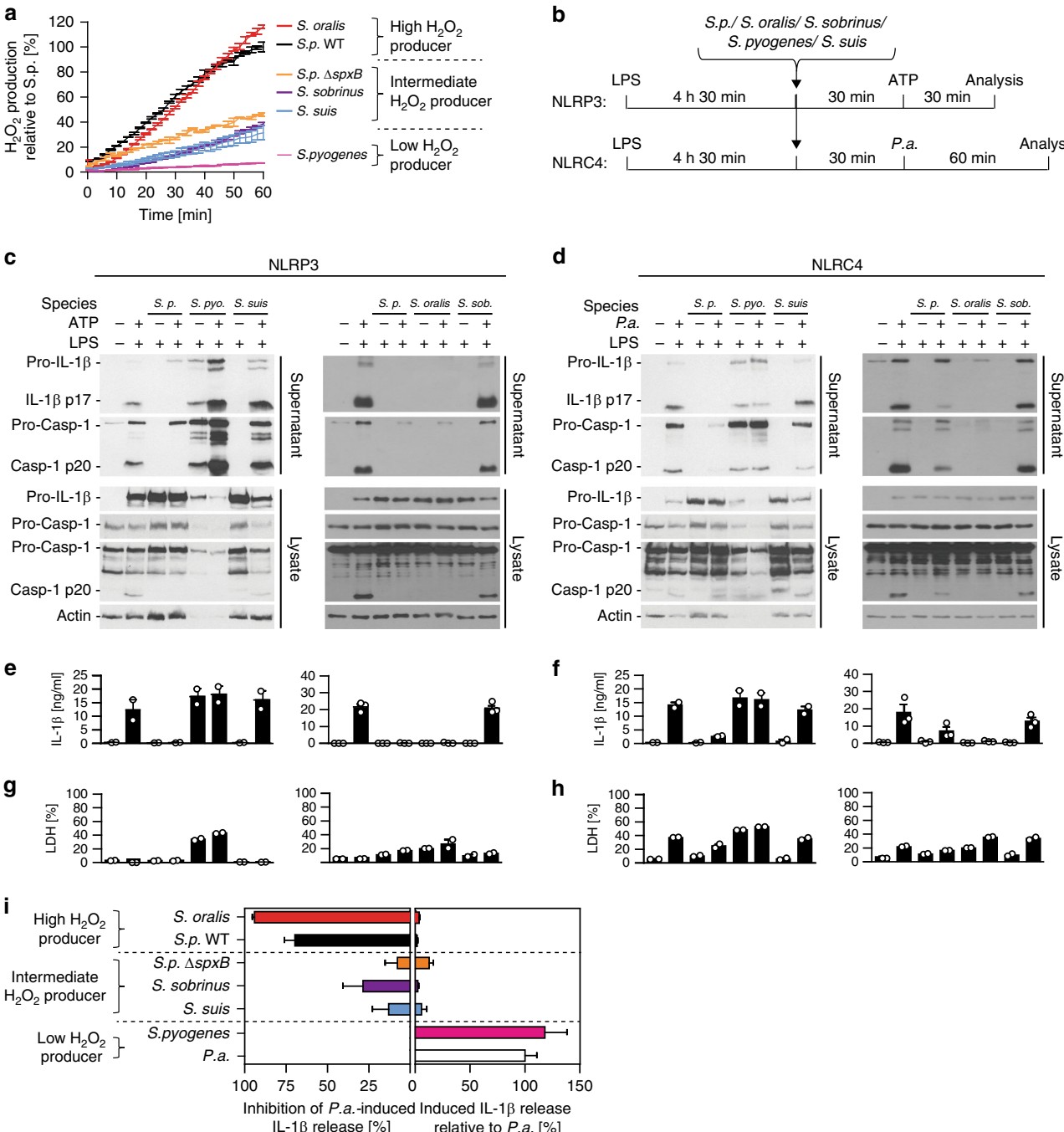

**Fig. 5** High H$_2$O$_2$-producing bacteria block inflammasome activation whereas low producers do not. **a** H$_2$O$_2$ production by *S. oralis*, *S. pneumoniae* Δ*spxB* (D39), *S. sorbnius*, S. suis, *S. pyogenes* relative to that by *S. pneumoniae* WT (D39) (100%). Data are representative of two independent experiments performed in duplicates or triplicates presented as the mean ± s.d. **b** Schematic diagrams of experimental setups used in (**c–h**). **c**, **d** Immunoblot analysis of Caspase-1 and IL-1β processing, **e**, **f** ELISA analysis of corresponding supernatants for IL-1β secretion and **g**, **h** LDH release of LPS-primed BMDMs pre-treated with *S. pneumoniae* D39 WT (*S. p.*), *S. pyogenes* (*S. pyo.*), *S. suis*, *S. oralis* or *S. sorbinus* (*S. sob.*) (MOI 40) for 30 min before **c**, **e**, **g** stimulation with ATP (5 mM) for 30 min or **d**, **f**, **h** infection with *P. aeruginosa* (MOI 20) for 60 min. Data in **c** and **d** are representative of two or three independent experiments, respectively. ELISA and LDH release data are from two or three independent experiments and are depicted as the mean ± s.e.m. **i** Relative inflammasome inhibition or activation by *Streptococcus* species presented as the mean ± s.e.m. Inhibition depicts decrease in *P.a.*-induced IL-1β release during co-infection with indicated bacteria. Activation depicts IL-1β release by indicated bacteria relative to that by *P.a.*. Data are presented as the mean ± s. e.m. of 2–9 experiments. Source data are provided as a Source Data file

an unexpected role of H$_2$O$_2$ in the negative regulation of inflammasomes.

Inflammasomes are vital players in the eradication of bacterial pathogens[23,25,26]. Correspondingly, recent studies also reveal that an increasing number of pathogens are equipped with

mechanisms to circumvent inflammasome activation[13,14,46]. Such pathogens can overcome inflammasome-dependent host defenses either by evading receptor recognition or through active mechanisms of inflammasome suppression[14]. The avoidance strategy is a stealth approach whereby pathogens disguise their

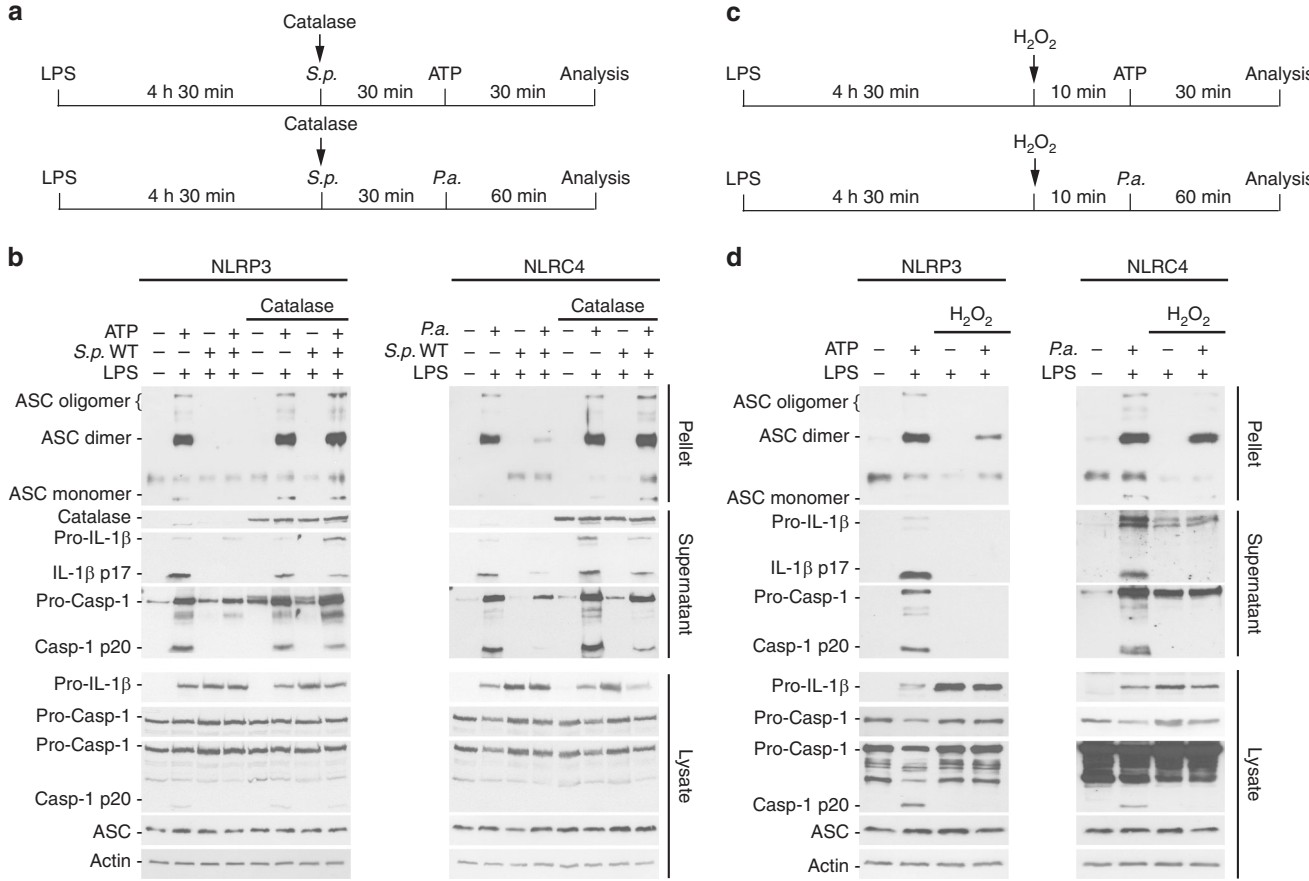

**Fig. 6** *S. pneumoniae* impairs inflammasome activation *via* SpxB-mediated H$_2$O$_2$ release. **a** Schematic diagrams of the experimental setups used in (**b**). **b** ASC oligomers, Caspase-1 and IL-1β processing in LPS-primed BMDMs pre-treated with *S.p.* WT (D39, MOI 100) in the presence of catalase (100 U mL$^{-1}$) for 30 min before ATP stimulation (5 mM, 30 min) or *P.a.* infection (MOI 20, 60 min). **c** Schematic diagrams of the experimental setups used in **d**. **d** ASC oligomers, Caspase-1 and IL-1β processing in BMDMs pre-treated with 50 µM H$_2$O$_2$ for 10 min prior to stimulation with ATP (5 mM, 30 min) or infection with *P.a.* (60 min, MOI 20). The data in **b** and **d** are representative of three independent experiments. Source data are provided as a Source Data file

presence from the host for example by restricting the bioavailability of inflammasome stimuli[47–50]. Active suppression on the other hand involves targeted modulation of inflammasomes by virulence factors that interact with host cell proteins[28,51–54]. The active mechanisms of inflammasome suppression reported thus far seem to require an intimate host-pathogen contact to permit the delivery of inhibitory effectors into the host cells[13,14,46]. Arguably, such directed delivery of virulence factors is probably essential to ensure that immune suppression is not global but only limited to the infected and not the healthy bystander cells. In this study, we report a new mechanism of inflammasome suppression that does not require direct bacterial contact with or delivery of bacterial proteins into host cells. H$_2$O$_2$ is a diffusible small molecule that easily crosses biological membranes and can modulate the activity of different cellular proteins. We show that by releasing H$_2$O$_2$ into their surroundings, bacteria such as *S. pneumoniae* and *S. oralis* cause oxidative stress in host cells and that this results in suppression of different inflammasome pathways.

The demonstration that bacteria-derived H$_2$O$_2$ inhibits inflammasomes and that this is likely a common mechanism employed by diverse bacterial species to counter the innate immune system is somewhat unexpected. Previously, several studies have linked reactive oxygen species (ROS) with the activation of the NLRP3 inflammasome[55–58]. However, other studies have indicated that ROS do not trigger inflammasomes per se but support the priming phase of NLRP3 activation[25,59–61]. In fact,

curiously and in agreement with the conclusions herein, it has been observed that genetic defects in ROS generation do not result in diminished inflammasome activation but rather in enhanced inflammasome activation[25,61,62]. On the other hand excessive generation of ROS and reactive nitrogen species by immune cells have been linked to diminished inflammasome activation[25,43,63,64].

Oxidative stress is known to result in oxidative damage of many cellular proteins[65,66]. The data herein show that ASC oligomerization—a central node for many inflammasome pathways—is highly sensitive to oxidative inhibition. Therefore, as previously discussed[6], although transient ROS generation might be essential for some steps such as priming, our findings support the view that oxidative stress is largely unfavourable for inflammasome activation.

Our understanding of host-microbe interactions has mostly been gleaned from studying monomicrobial infections. However, many infections such as those of the respiratory system are polymicrobial[16,67]. In these infections, one microbe predisposes the host to colonisation by others. *S. pneumoniae* possibly contributes to the aetiology of polymicrobial lung colonization[16,17]. The results here show that H$_2$O$_2$-mediated inflammasome inhibition not only enables bacteria to avoid invoking strong self-limiting inflammatory responses, but also to block the responsiveness of immune cells to other danger signals in co-infection setting. Thus, beyond demonstrating how *S. pneumoniae* and other bacteria subvert the innate

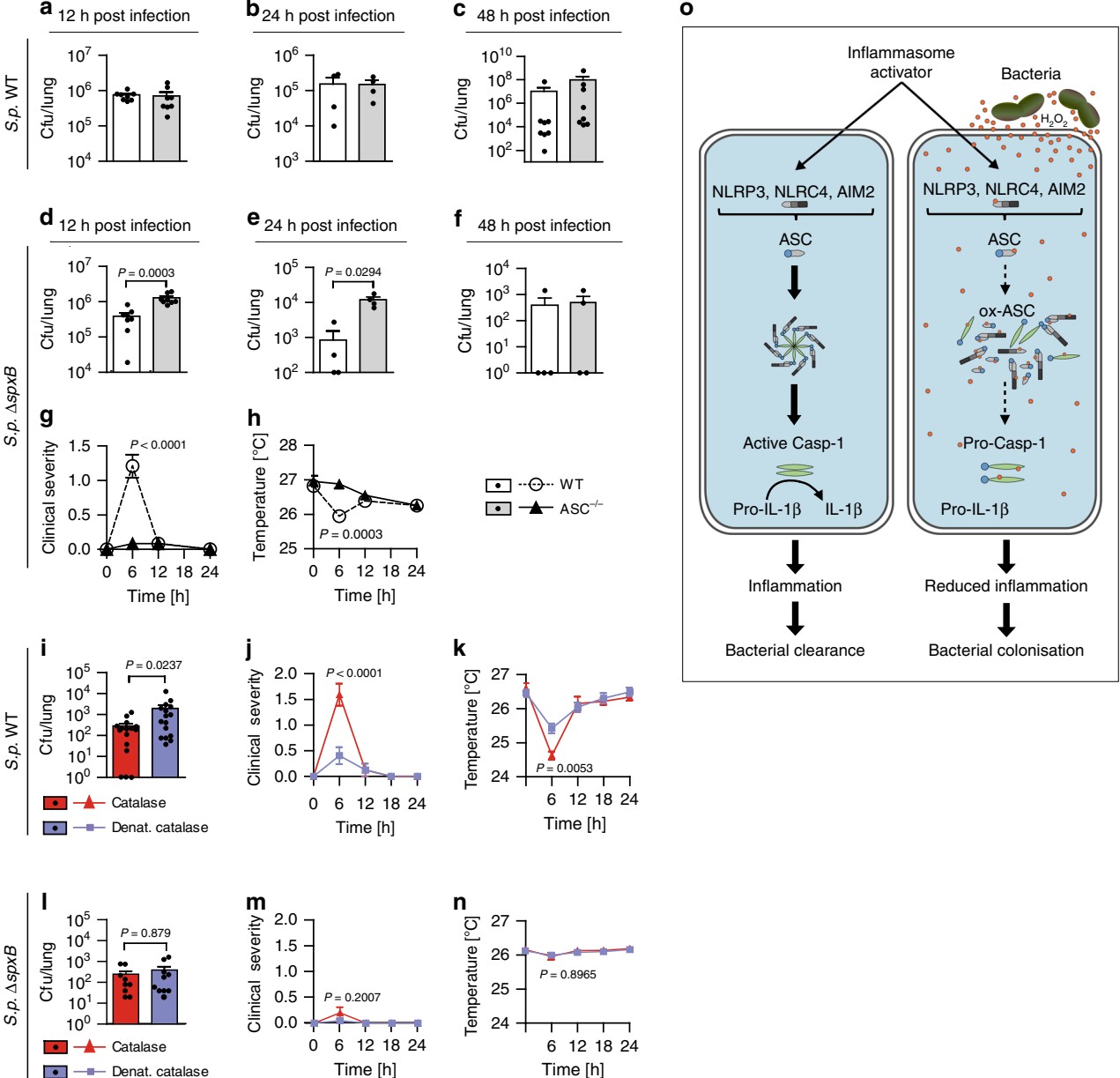

**Fig. 7** $H_2O_2$ release by *S. pneumoniae* suppresses inflammasome-dependent bacterial clearance. **a**–**c** *S.p.* WT (D39) burden (cfu) in the lungs of WT and ASC$^{-/-}$ mice **a** 12 h, **b** 24 h, and **c** 48 h after intranasal infection (1–2 × 10$^7$ cfu/mouse). (**d**–**f**) *S.p.* Δ*spxB* (D39) burden in the lungs of WT and ASC$^{-/-}$ mice **d** 12 h, **e** 24 h, and **f** 48 h post infection (1–2 × 10$^7$ cfu/mouse). **g** Clinical severity and **h** subcutaneous body temperatures of WT and *ASC$^{-/-}$* mice after *S.p.* Δ*spxB* (D39) infection. Results in **a** to **h** are representative of 2 independent experiments with a total of 4–8 animals per group (1–2 × 10$^7$ cfu/mouse). **i** Bacterial burden, **j** clinical severity, and **k** subcutaneous body temperature of WT mice co-inoculated with *S.p.* WT (D39) and active or denatured catalase (500 U per mouse) for 24 h. Results are representative of three independent experiments with a total of 16 animals per group (1 × 10$^7$ cfu/mouse). **l** Bacterial burden, **m** clinical severity, and **n** subcutaneous body temperature of WT mice co-inoculated with *S.p.* Δ*spxB* (D39) and active or denatured catalase (500 U per mouse) for 24 h. Results are representative of two independent experiments with a total of 10 animals per group (1 × 10$^7$ cfu/mouse). The data in **a** to **n** are shown as mean ± s.e.m. **a**–**f**, **i**, **l** *P* values determined by Mann Whitney test. **g**, **h**, **j**, **k**, **m**, **n** *P* values determined by two-way ANOVA followed by Bonferroni post-test. **o** Proposed model of inflammasome inactivation by high $H_2O_2$-producing bacteria. Source data are provided as a Source Data file

immune system, this work opens up possibilities for further studies to better understand the aetiology of polymicrobial host colonization.

Finally, the demonstration that $H_2O_2$ inhibits inflammasomes advances the current knowledge beyond the conventional view of ROS as drivers of inflammation[7,68] and highlights an under-appreciated anti-inflammatory effect of ROS. The latter notion is consistent with increasing clinical and experimental evidence linking a reduction in intracellular ROS with hyper-inflammation[61–63,69].

## Methods

**Ethics statement**. Experiments involving humans were performed according to the recommendations of the local Research Ethics Committee of Umeå University (Regionala etikprövningsnämnden i Umeå), as approved in permit Dnr 2012-501-31M. Full informed consent was obtained from donors in compliance with the Declaration of Helsinki. Animal experiments were carried out according to the guidelines set out by the Umeå Regional Animal Ethic Committee (Umeå Regionala Djurförsöksetiska Nämnd), Approval no. A53-14.

**Mice**. All mice in this study were on C57BL/6 background. *Asc$^{-/-}$* mice[27] were obtained from Genentech, South San Francisco, USA. All mice were bred at the

Umeå Transgene Core Facility and in vivo infection experiments were performed at the Umeå Centre for Comparative Biology. Mice were maintained under specific pathogen free conditions.

**Animal infections**. Age- and sex-matched adult mice (8–20 weeks old) were anaesthetized using isoflurane with the XGI-8 gas anaesthesia system (Caliper), inoculated intranasally with 30 µl of a suspension containing $1–2 \times 10^7$ cfu of *S. pneumoniae* strain D39 wild type or D39 Δ*spxB* and then suspended by their teeth for 10 min to enable the infection inoculum to descend into the lungs. At different time points after infection, mice were monitored for clinical symptoms such as weight loss and temperature. Clinical severity scoring was based on an arbitrary scale from 1 to 4 where 1 represented mice with mild but visible symptoms such as slowed activity whereas 4 represented those with highest morbidity, i.e. with a combination of hunchback posture, lethargy, loose fecal pellet, ruffled fur, >25% weight loss and difficulties in breathing and movement and hence had to be euthanized. Bacterial burden in the lung was determined by plating lung homogenates on blood agar plates to determine the cfu. For in vivo analysis of cytokines, the lungs were gently disrupted in PBS to release cells from the connective tissue. Lung flushings containing extracellular fluids and cell pellets were separated by centrifugation, and extracellular fluids were analysed by ELISA.

**Bacterial strains and growth conditions**. *S. pneumoniae* D39 wild-type and D39 Δ*spxB* strains were provided by Jetta Bijlsma[45] (MSD animal health, Netherlands). TIGR4 Serotype 4 (BAA-334) and R6 Serotype 2 (BAA-255) were obtained from ATCC. The *S. pneumoniae* SpxB-complemented D39 Δ*spxB* + SpxB (P878 + pMU1328::300 + *spxB*: P1221) strain and the control strain D39 Δ*spxB* (D39 spxB:: Tn*phoA*(erm): P878), R6x and R6x Δ*spxB* (R6x *spxB*::Tn*phoA*(erm): P1167)[9,19] were from Jeffrey Weiser (NYU School of Medicine, New York, USA). *S. pneumoniae* D39 wild-type and D39 Δ*ply* strains were provided by Sven Hammerschmidt (Universtiy of Greifswald, Germany). *Salmonella enterica* serovar Typhimurium strain SL1344 was provided by Siegfried Weiss (Helmholtz Centre for Infection Research, Braunschweig, Germany). *P. aeruginosa*-Xen41 and *S. pneumoniae*-Xen10 strain A66.1 Serotype 3 were obtained from Caliper. *Streptococcus pyogenes* NZ131 (serotype M49) was provided by Victor Nizet (University of California, USA). *Streptococcus suis* (serotype 2) strain 10 was provided by Peter Valentin-Weigand (Tierärztliche Hochschule Hannover, Germany). *Streptococcus oralis* ATCC 35037 and *Streptococcus sobrinus* CCUG 21019 were provided by Jan Oscarsson (Umeå University, Umeå, Sweden). All bacterial strains used in this study were grown to exponential phase for infection experiments. *Streptococcus* species were incubated on blood agar plates overnight at 37 °C and 5% CO₂ followed by inoculation and growth in Brain Heart Infusion broth at 37 °C and 5% CO₂ for 2 1/2 h; *S.* Typhimurium and *P. aeruginosa* were incubated for 24 h at 37 °C on LB-agar (Lennox), then inoculated and grown overnight in Luria broth at 37 °C at 150 rpm to stationary phase followed by subculture in Luria broth at 37 °C at 150 rpm for 2 h to exponential phase.@@@

**ROS measurements**. $H_2O_2$ production by the different *Streptococcus* species was determined by adding 50 µM luminol and 1.2 U mL⁻¹ horseradish peroxidase (HRP) to 100 µl of pre-warmed OptiMEM at 37 °C containing $1.25 \times 10^5$ bacteria. The $H_2O_2$ levels, expressed as relative light units (RLU), were measured every 2 min for 1.5 h at 37 °C using a Tecan Infinite M200 plate reader.

The $H_2O_2$ amounts produced by *S.p.* D39 WT, *S.p.* D39 Δ*spxB*, and *S.p.* D39 Δ*spxB* + SpxB were determined by the Fluorometric Hydrogen Peroxide Assay Kit from Sigma according to the manufacturers' instructions. $1.5 \times 10^6$ BMDMs in 1 ml OptiMEM were infected with *S.p.* at MOI 1, 10, or 100 in the presence (or not) of 100 U mL⁻¹ catalase for 1.5 h. Cell culture supernatants were collected, bacteria removed by centrifugation at $3000 \times g$ for 2 min, and directly subjected to $H_2O_2$ quantification.

**Cell culture, infection and stimulation**. BMDM differentiation was performed by culturing bone marrow cells in 20% conditioned L-929 culture medium for 5 days. Human PBMCs were isolated from peripheral blood using Ficoll-Paque PLUS. To measure inflammasome responses, cells were primed with 500 ng mL⁻¹ LPS for 4.5 h and then infected with the indicated *Streptococcus* species for 30 min at the indicated multiplicity of infection (MOI). Thereafter, cells were further stimulated with 5 mM ATP or 1 µM nigericin for 30 min, or infected with *P. aeruginosa* Xen41 at MOI 20 or *S.* Typhimurium SL1344 at MOI 10 for 60 min. In all cases, the bacteria were centrifuged onto BMDMs at $250 \times g$ for 5 min to ensure comparable cell contact. 100 U mL⁻¹ catalase (except where indicated otherwise) was added simultaneously with the indicated *Streptococcus* species to the cell culture medium, while 50 µM $H_2O_2$ was added 10 min before the addition of inflammasome −activators. For infection experiments, when cells were incubated for several hours (6–12 h), cells were infected for 60 min with *S. pneumoniae*, while for −further incubation the cell culture medium was replaced by medium containing 25 µg mL⁻¹ gentamicin.

**Calculation of bacterial uptake**. LPS-primed BMDMs for 4.5 h were infected (or not) with *S. pneumoniae* WT at MOI 100 for 30 min. Cells were then infected with *P. aeruginosa* at MOI 20 for 1 h. Infections were synchronised by 5 min

centrifugation at $250 \times g$. Cells were washed three times with PBS and then lysed in PBS containing 1% Triton X-100 and serial dilutions were plated on tetracycline-containing LA plates.

**Cytokine ELISAs and LDH assay**. The BD OptEIA Mouse IL-1β ELISA Set and BD OptEIA Human IL-1β ELISA Set (BD Biosciences), Mouse IL-18 ELISA Kit (MBL International), TNF alpha Human ELISA Kit (abcam) and Mouse TNF-α DuoSet (R&D Systems) were used according to the manufacturers' instructions. LDH release was determined by CytoTox-ONE Homogeneous Membrane Integrity Assay (Promega) according to the manufacturers' instructions.

**Immunoblotting analysis**. Supernatants from BMDMs ($1.5 \times 10^6$/well) maintained in serum-reduced medium (OptiMEM, Invitrogen) during stimulation were collected. Cell debris was removed by centrifugation, and proteins were precipitated by methanol-chloroform extraction. The precipitates were resuspended in $2 \times$ Laemmli buffer, and the cells were directly lysed in $2 \times$ Laemmli buffer. Proteins were separated on 13.5% SDS-PAGE gels and immunoblotted onto nitrocellulose membranes (Amersham). Membranes were blocked for 1 h in $1 \times$ Roti Block (Roth) and subsequently incubated with different primary antibodies overnight. After incubation with HRP-labelled secondary antibodies, the proteins were detected using ECL substrate and X ray films. All uncropped Western blots are provided as part of the Source Data file.

**Analysis of carbonylated proteins**. Equal amounts of total protein lysates in 6% SDS in 20 mM Tris-HCl, pH 7.4, and 150 mM NaCl were treated with 5 µM 2,4-dinitrophenylhydrazine (DNPH) in 5% trifluoroacetic acid (TFA) for 15 min. Negative controls were treated with 5% TFA without DNPH. Cell lysates were neutralised by adding 2 M Tris, pH > 11, mixed with $2 \times$ Laemmli buffer and analysed for DNP by immunoblotting. Alternatively, the cells were lysed in 20 mM Tris-HCl, pH 7.4, 150 mM NaCl, 1 mM EDTA, 1% NP-40, 0.5% deoxycholic acid, and 0.2% SDS containing a protease inhibitor cocktail (Roche). Protein A/G PLUS beads (Santa Cruz) with bound anti-ASC (or rat IgG control) were added to the lysates, and the mixture was incubated overnight at 4 °C. The immunoprecipitates were collected, washed, resuspended in 6% SDS in 20 mM Tris, pH 7.6, with 137 mM NaCl and heated for 5 min at 70 °C. The immunoprecipitated proteins were derivatized with 10 mM DNPH in 5% TFA for 15 min and neutralised using 2 M Tris, pH > 11, and buffered with $6 \times$ Laemmli buffer. Then the proteins were analysed for DNP by immunoblotting.

**ASC oligomerization**. ASC oligomerization was assayed in LPS-primed cells treated with inflammasome activators. $3 \times 10^6$ cells were lysed in 20 mM Hepes, pH 7.5, 150 mM KCl, and 1% NP-40 with a protease inhibitor tablet (Roche) and sheared using a 21G×2" needle. The cell lysates were centrifuged at $3300 \times g$ for 10 min at 4 °C, and the pellets were washed twice with 1 ml of PBS. This was followed by incubation in 2 mM suberic acid bis(N-hydroxy succinimide ester) (DSS) for 30 min at RT. After centrifugation at $3300 \times g$ for 10 min at 4 °C, the supernatants were removed and the resulting pellets were dissolved in $2 \times$ Laemmli buffer. The proteins were separated on 10% SDS-PAGE gels and analysed by immunoblotting as described before.

**Microscopy**. For imaging of active Caspase-1, BMDMs were seeded on glass coverslips in 24-well plates at a density of $2.5 \times 10^5$ cells/well. LPS-primed cells were treated with FAM-YVAD-FMK FLICA reagent in DMSO or DMSO only for 15 min, followed by inflammasome activation. Afterwards, the cells were washed with PBS, fixed in 4% (w/v) paraformaldehyde and incubated with the rat anti-ASC (1:500) primary antibody in blocking buffer (3% (w/v) BSA and 0.1% (w/v) saponin) for 30 min. After washing, the cells were incubated with an AlexaFluor 546-conjugated secondary antibody (1:250) for 30 min, stained with DAPI before mounting and imaged using a Nikon Eclipse C1 Plus Eclipse confocal microscope. To quantify Caspase-1 activity, random image sections with at least 100 cells were counted for each condition.

**Reagents and antibodies**. Adenosine 5′-triphosphate disodium salt hydrate (ATP), nigericin sodium salt, bovine catalase (C-40), gentamicin, suberic acid bis (N-hydroxy succinimide ester) (DSS), 2,4-dinitrophenylhydrazine (DNPH) and monoclonal anti-β-Actin (clone AC-74, #A2228; dilution 1:10,000) were purchased from Sigma-Aldrich. Catalase in in vivo experiments was from Worthington Biochemical Corporation. Lipopolysaccharide (LPS-EB, TLR4/TLR2 ligand) was from InvivoGen. Listeriolysin O (LLO) was purified from supernatants of non-pathogenic *Listeria innocua* hyperexpressing listeriolysin O as described previously[70]. $H_2O_2$ was from Merck-Millipore. Rat anti-mouse Caspase-1 p20 (Clone 4B4.2.1; used dilution 1:2,000) and rat anti-mouse ASC (CARD5) (Clone 8E4.1; for immunoblotting: dilution 1:2,000, for immunoprecipitation: 2 µg per 500 µg cell lysate, for immunofluorescence staining: dilution 1:500) were obtained from Genentech, San Francisco USA. Goat anti-IL-1β was from R&D Systems (#AF-401-MA; dilution 1:2,500). NLRP3 rabbit monoclonal antibody (D4D8T) was from Cell Signaling Technology (#15101; dilution 1:1,000). Polyclonal rabbit anti-dinitrophenol (DNP) was from Abcam (#AB6306; dilution 1:1,000). Anti-HRP-

labelled secondary anti-mouse and anti-rabbit were from GE Healthcare Amersham (#NA931; dilution 1:10,000) and Cell Signaling Technology (#7074; dilution 1:10,000), respectively. The anti-HRP-labelled secondary anti-goat and anti-rat antibodies (#sc-2020 and sc-2006; dilution 1:10,000) and normal rat-IgG control (#sc-2026; used for immunoprecipitation: 2 μg per 500 μg cell lysate) were from Santa Cruz Biotechnology. The AlexaFluor 546 goat anti-rat IgG secondary antibody was from Life Technologies (#A11081; dilution 1:250). ECL substrates and X-ray films were purchased from GE Healthcare Amersham (Hyperfilm ECL). FAM-YVAD-FMK FLICA (dilution 1:150) was from ImmunoChemistry Technologies. Ficoll-Paque PLUS was from GE Healthcare.

**Statistical information**. The data in the text and Figures are expressed as the mean with the standard error of the mean (±s.e.m.) or if representative measurements of technical replicates are shown as mean with standard deviation (±s.d.). Statistical comparisons were done using either a one–way ANOVA or two-way ANOVA with Bonferroni post-test or Mann Whitney test. $P < 0.05$ was considered as statistically significant. The survival rates were analysed by the Kaplan-Meier method and $P$ values were determined by Gehan-Breslow-Wilcoxon test.

**Reporting summary**. Further information on research design is available in the Nature Research Reporting Summary linked to this article.

## Data availability
All data are available in the manuscript and its Supplementary Information files. Raw source data for Figs. 1–7 and Supplementary Figs. 1–10 are presented in the Source Data file.

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

## Acknowledgements

We are grateful to Jetta J. E. Bijlsma of the University of Groningen, Groningen, Netherlands for providing D39 *ΔspxB* strain, and to Jeffrey Weiser, New York University School of Medicine, USA for the *S. pneumoniae* SpxB-complemented strain (D39 *ΔspxB* + *SpxB*) and others. We thank, Victor Nizet (University of California, USA) for *Streptococcus pyogenes* NZ131 (serotype M49), Peter Valentin-Weigand (Tierärztliche Hochschule Hannover, Germany) for *Streptococcus suis* serotype 2 strain 10 and Jan Oscarsson (Umeå University, Sweden) for *Streptococcus oralis* ATCC 35037 and *Streptococcus sobrinus* CCUG 21019. This work was supported by funding from the Laboratory for Molecular Infection Medicine Sweden (MIMS). This work was supported by Umeå University, Stockholm University and Swedish Research Council grants (reference numbers 2015-02857 and 2016-00890) to N.O.G.

## Author contributions

N.O.G. conceived, designed and supervised the study. S.F.E. conceived, designed and performed experiments. S.F.E. and N.O.G. prepared the manuscript.

## Additional information

**Competing interests:** The authors declare no competing interests.

