## [Peer Review File · Nature Communications]

Reviewers' comments:

Reviewer #1 (Remarks to the Author):

In the manuscript "Hydrogen peroxide release by bacteria suppresses inflammasome-dependent innate immunity", Erttmann and Gekara report a novel mechanism in which H₂O₂ production by *Streptococcus pneumoniae* can inhibit Nlrp3, Nlrc4 and Aim2 inflammasome signaling pathways via ASC oxidation to mitigate anti-microbial innate immune defenses. Given the fact that ROS generation is generally regarded by the field as an activator of the Nlrp3 pathway rather than an inhibitor, this study can have a considerable impact on the way we think about ROS in anti-bacterial defenses. However, as this study provides a concept that is counterintuitive for many researchers in this field it needs very convincing data supporting the novel concept. In general, I think this study provides quite some convincing in vitro data that support the mechanism in which SpxB produces H₂O₂ causing inactivation of inflammasomes in macrophages. However, I think there is one very obvious in vitro experiment that is lacking and that would provide great support for their concept. In addition, I think the in vivo data provided by the authors are a bit less clear and could use some additional clarification. Below are my specific comments.

Major comments

1. The authors show in Sup Fig 1a-c that *Streptococcus pneumoniae* is only a weak activator of inflammasome responses. They argue that this is due to the capacity of this bacterium to produce H₂O₂ causing inactivation of inflammasomes, a concept that they explain in their study in a fairly convincing manner by using these bacteria to inhibit inflammasome activation triggered by independent activators. To make their point, the authors use SpxB mutant bacteria that cannot produce H₂O₂. Indeed, these SpxB mutants do not inhibit Nlrp3 and Nlrc4 inflammasomes as WT bacteria do. However, one obviously expected consequence of blocking H₂O₂ in the SpxB mutants would be that this mutant by itself is a better inflammasome activator than the WT bacteria. Experimental settings comparing these bacteria were done e.g. in Fig 3b, but probably the time frame of bacterial stimulation was not long enough to see activation. Can the authors compare the WT and SpxB mutant bacteria side by side in inflammasome activation experiments as shown in Sup Fig 1a-c? Enhanced or more rapid activation of inflammasome responses by the SpxB mutant would provide great support for H₂O₂ as inhibitor of the endogenous bacterial inflammasome triggering.
2. Related to the above comment, it is strange that catalase treatment experiments such as in Fig 6b do not lead to inflammasome activation by the WT *Streptococcus pneumoniae* (lane 7 in western blots, LPS + Sp + catalase). As suggested for the SpxB mutants in the above comment, blocking H₂O₂ in the WT bacteria by catalase would be expected to relieve the inhibition of inflammasome activation and thus to inflammasome responses to these WT bacteria. Perhaps in Fig 6b the time of bacterial triggering was not long enough to see this effect? Can the authors also include catalase treatments of WT bacteria in the above suggested experiment as additional support for H₂O₂ as inhibitor of the endogenous bacterial inflammasome triggering? If these experiments do not support this notion, then one may wonder what the actual in vivo inflammasome trigger is upon *Streptococcus pneumoniae* infection. Does H₂O₂ inhibit bacterial inflammasome activation in vivo, or does it inhibit inflammasome activation by other triggers?
3. Based on Fig 1g-h the others claim that *Streptococcus pneumoniae* infection with the SpxB mutants is cleared by inflammasome activation based on their observation that IL-1b but not TNF is upregulated in the lungs. However, although only the IL-1b increase is statistically significant, both IL-1b and TNF show similar upward trends. I would be a bit more careful with the statement that the stronger innate immune response to the mutant bacteria is inflammasome-dependent (lines 65-67).
4. In Fig 7i-k it would be good to include the SpxB mutant bacteria in the experiment. The catalase provided intranasally might work also on the host immune system in addition to acting on the bacteria. In order to show that the adverse effects are due to blocking H₂O₂ from the bacteria and not due to effects on the immune system, SpxB mutants should be included.

5. The authors put quite some effort in trying to convince the reader that the effects of H₂O₂ lie in the activation step rather than the priming step of inflammasome responses. Judging from the pro-IL1 β levels in the western blots I would agree with the authors that the effect is in the activation step (this is also consistent with the effect on ASC oligomerization). However, the authors frequently point to TNF production as a measure for judging inflammasome priming, whereas I think this makes little sense. For instance, the argument for TNF staying the same in Fig 2k-m does not make a lot of sense, since the TNF measured there presumably derives from stimulating the cells with LPS and TLR stimulation by the *S. pneumoniae*. ATP and nigericin themselves do not induce TNF, and *P. aeruginosa* stimulation does not increase the TNF levels that were induced before by the LPS and *S. pneumoniae*. Also the experiments in Sup Fig 2 use TNF secretion as a proxy for inflammasome priming but in the end make little sense. I would invite the authors to think carefully what is needed to address this question. In my opinion, pro-IL-1 β levels and the fact that the inhibition occurs already at the level of ASC oligomerization is sufficient proof.

Minor comments

1. Streptococci have been reported to activate NLRP3, AIM2 and NLRC4 inflammasomes. Although the authors show an inhibitory effect of *S. pneumoniae* WT upon poly(dA:dT) stimulation, they do not follow up on the AIM2 inflammasome signaling thereafter. This experiment seems to be a bit out of the focus of the story. I would suggest to either omit it, or to further explore whether the inhibitory mechanism is the same for AIM2 as in NLRP3 and NLRC4 inflammasomes?
2. Sup Fig1a: I guess the Y-axis of the IL1 β plot should be pg/ml instead of ng/ml.
3. Sup Fig6: Please align the figure labels a bit better.
4. There are quite some spelling errors in the manuscript, e.g. 'Age- anf sex-matched' on line 253.

Reviewer #2 (Remarks to the Author):

In this study the authors show that while H₂O₂ can facilitate inflammasome assembly, "massive" amounts of H₂O₂ can dampen inflammasome activation. They demonstrate this in the context of infection with *S pneumoniae*, a pathogen that lacks catalase. The authors show that this phenotype can be reversed by the addition of exogenous catalase both in vitro and in mice. The phenotype is absent in sbx (a pyruvate oxidase) -deficient bacteria correlating with decreased H₂O₂ release in this strain.

This observation is of interest and the experiments presented are overall well-controlled and convincing. I have a few comments to improve the physiological relevance of the study and data interpretation.

- 1) The mechanism of H₂O₂ inhibition is not addressed in the manuscript. The authors suggest in figure S6 that high level of H₂O₂ may oxidize inflammasome components but they do not demonstrate that this may impact activity.
- 2) While the authors show that assembly of inflammasomes is impaired by high doses of H₂O₂, they cannot exclude an effect on signal 1 (priming). As a control and to support the notion that H₂O₂ does not affect the inflammasome competence of the cell it would be important to complete the data in Fig 3 and 4 with expression levels for inflammasome platforms including NLRP3, AIM4 and NLRC4.
- 3) The authors should investigate apoptosis induction in presence of high levels of H₂O₂. Oxidative stress can promote apoptosis and apoptotic caspases can negatively regulate inflammasome

activity.

Minor point

In some of the figure the legend is not clear is. For example in Fig1 the color chart for panel b to e is unclear. The legend is missing for b or does not corresponds to the picture in e.

Reviewer #3 (Remarks to the Author):

The modulation of the host immune response by *Streptococcus pneumoniae* largely depends on virulence bacterial factors. Pneumolysin and the capsule are the best studied virulence factors interfering with phagocytosis and modulation of host cells and host response as well. Moreover, pneumococci are producers of hydrogen peroxide (H₂O₂) and the concentrations of H₂O₂ compete out other bacteria in the hot niche. Because H₂O₂ is released in the host environment this chemical compound is also thought to modulate the immune response. Here the authors show that H₂O₂ inhibits inflammasomes by using wild-type and isogenic *spxB* knockout strains. *SpxB* is a pyruvate oxidase and converts pyruvate under aerobic or microaerobic conditions into acetylphosphate, carbon dioxide and H₂O₂. Importantly, pneumococci do not produce a catalase as other Gram-positive or -negative bacteria, however, they are protected against extracellular oxidative stress. This point should be introduced more specifically, because this is a highly important molecular mechanism in the context of H₂O₂ production and resistance against ROS produced by immune cells. Similar to pneumococci other H₂O₂ producing bacteria block inflammasomes dependent on the amount of H₂O₂ released. The authors further validated their data by using *Asc*^{-/-} KO mice. ASC oligomerization is required for activation of the inflammasome complexes.

Taken together, the authors unraveled a novel mechanism of deactivation of inflammasomes and immune suppression probably facilitating colonization.

Although the study is well done and technically sound, the reviewer has a few important aspects that have to be addressed, probably also experimentally.

2. Specific comments and experimental design

- the introduction has been kept extremely focused and short. There are at least two aspects which have to be addressed: 1. how is the pneumococcus protected against H₂O₂ and 2. the amount of H₂O₂ released under in vitro conditions and under in vivo conditions
- there are other studies (e.g. Huet et al., 2017 Crit Care Med Protective Effect of Inflammasome Activation by Hydrogen Peroxide in a Mouse Model of Septic Shock; Reck Nunes et al 2018 (<https://doi.org/10.1016/j.preghy.2018.07.006>) which show that oxidative stress activates the inflammasome, thereby leading to elevated levels of IL-1β and TNF-α, enhanced gene expression of NLRP3, caspase 1 or that that increased concentrations of active caspase-1 and interleukin-1β are related to an increased concentration of hydrogen peroxide. These results should be cross-linked with the results gained in the submitted study
- page 5: BMDMs were infected with *S. pneumoniae* post stimulation with LPS and caspase-1 / IL1β processing were compared. What happens, when BMDMs were not stimulated with LPS?
- Because the strains are positive for pneumolysin: is there any relevance for pneumolysin which is released as well (despite being an intracellular toxin) pneumolysin is known to activate the NLRP3 inflammasome
- page 6: *S. pneumoniae* inhibits dose-dependently secretion of caspase-1 etc. Which amounts of H₂O₂ are needed. A kinetics would be nice and in addition, which is the amount or concentration that is needed at least.
- the authors have used also a complemented *spxB*-mutant. Are the levels of H₂O₂ production comparable to the wild-type?
- page 6 - in the experiments represented in Figure 4 the complemented strain should be used as well

- the *spxB* knockout showed a growth advantage via the wild-type. I assume this is in BHI, i.e. in a complex medium. How about growth in a minimal medium?
- Figure 1. the infection dose should be given in the figure legend;
- why is the *spxB*-mutant still producing H₂O₂ (Figure 1)?

Minor points:

- page 11, line 222-223: please check sentence and wording
- pge 12, line 253: Age-and-sex....(?: and?)
- page 13, line 269: the authors have to describe more precisely in their assays (results) and Figure legends which pneumococcal strain have been used. In addition, the R6 is uncapsulated and can't have any serotype.

Reviewer #1 (Remarks to the Author):

In the manuscript “Hydrogen peroxide release by bacteria suppresses inflammasome-dependent innate immunity”, Erttmann and Gekara report a novel mechanism in which H₂O₂ production by *Streptococcus pneumoniae* can inhibit Nlrp3, Nlrc4 and Aim2 inflammasome signaling pathways via ASC oxidation to mitigate anti-microbial innate immune defenses. Given the fact that ROS generation is generally regarded by the field as an activator of the Nlrp3 pathway rather than an inhibitor, this study can have a considerable impact on the way we think about ROS in anti-bacterial defenses. However, as this study provides a concept that is counterintuitive for many researchers in this field it needs very convincing data supporting the novel concept. In general, I think this study provides quite some convincing in vitro data that support the mechanism in which SpxB produces H₂O₂ causing inactivation of inflammasomes in macrophages. However, I think there is one very obvious in vitro experiment that is lacking and that would provide great support for their concept. In addition, I think the in vivo data provided by the authors are a bit less clear and could use some additional clarification. Below are my specific comments.

Response: We thank the reviewer for these supportive comments and suggestions for improvements. We have now included all the suggested additional experiments.

Comment #1: The authors show in Sup Fig 1a-c that *Streptococcus pneumoniae* is only a weak activator of inflammasome responses. They argue that this is due to the capacity of this bacterium to produce H₂O₂ causing inactivation of inflammasomes, a concept that they explain in their study in a fairly convincing manner by using these bacteria to inhibit inflammasome activation triggered by independent activators. To make their point, the authors use SpxB mutant bacteria that cannot produce H₂O₂. Indeed, these SpxB mutants do not inhibit Nlrp3 and Nlrc4 inflammasomes as WT bacteria do. However, one obviously expected consequence of blocking H₂O₂ in the SpxB mutants would be that this mutant by itself is a better inflammasome activator than the WT bacteria. Experimental settings comparing these bacteria were done e.g. in Fig 3b, but probably the time frame of bacterial stimulation was not long enough to see activation. Can the authors compare the WT and SpxB mutant bacteria side by side in inflammasome activation experiments as shown in Sup Fig1a-c? Enhanced or more rapid activation of inflammasome responses by the SpxB mutant would provide great support for H₂O₂ as inhibitor of the endogenous bacterial inflammasome triggering.

Response: We thank the reviewer for this thoughtful comment. We have now compared inflammasome activation in macrophages infected with *S.p.* WT and *S.p.* Δ spxB for 12 hours. Because of limited space we have only embedded these data here below (**Figure I**). Although inflammasome activation under these conditions was very weak, as we observed before (**Supplementary Fig. 1c**), Caspase-1 processing induced by *S.p.* Δ spxB was slightly stronger than that by *S.p.* WT.

Figure I Effect of SpxB on inflammasome activation by *S. pneumoniae*. Caspase-1 processing by BMDMs primed with 500 ng/ml LPS for 4.5 h then infected with *S. pneumoniae* D39 WT or Δ spxB at MOI 50, 100 or 200 for 12 h. Data are representative of two independent experiments.

Comment #2: Related to the above comment, it is strange that catalase treatment experiments such as in Fig 6b do not lead to inflammasome activation by the WT *Streptococcus pneumoniae* (lane 7 in western blots, LPS + Sp + catalase). As suggested for

the SpxB mutants in the above comment, blocking H₂O₂ in the WT bacteria by catalase would be expected to relieve the inhibition of inflammasome activation and thus to inflammasome responses to these WT bacteria. Perhaps in Fig 6b the time of bacterial triggering was not long enough to see this effect? Can the authors also include catalase treatments of WT bacteria in the above suggested experiment as additional support for H₂O₂ as inhibitor of the endogenous bacterial inflammasome triggering? If these experiments do not support this notion, then one may wonder what the actual in vivo inflammasome trigger is upon *Streptococcus pneumoniae* infection. Does H₂O₂ inhibit bacterial inflammasome activation in vivo, or does it inhibit inflammasome activation by other triggers?

Response: We thank the reviewer for this comment and have performed the suggested experiment. As noted by the reviewer, short (1 h) infections with *S.p.* WT do not induce inflammasome activation in the presence or absence of catalase. However, longer (12 h) infections led to mild inflammasome activation, which was increased by addition of catalase (**Supplementary Fig. 3f**). Our data in **Supplementary Fig. 3a** show that a major factor of inflammasome activation by *S. pneumoniae* is the thiol-dependent toxin pneumolysin (PLY). Therefore we also analysed inflammasome activation in response to the PLY homologue, listeriolysin O (LLO). Indeed, inflammasome activation by LLO was enhanced by catalase (**Supplementary Fig. 3e**) but impeded by H₂O₂-producing *S.p.* WT but not *S.p.* Δ *spxB* (**Supplementary Fig. 3b-d**).

To the reviewer's question: "... what is the actual in vivo inflammasome trigger upon *Streptococcus pneumoniae* infection?" We conclude that inflammasome activation by *S. pneumoniae* at early stages of infection is triggered indirectly by endogenous danger molecules from damaged tissue such as ATP; however, this inflammasome activation seems to be restrained by high amounts of H₂O₂. In addition to these danger molecules, at later stages of infection, bacterial components such as PLY contribute further to inflammasome activation. The cumulative effect is consistent with inflammasome-dependent clearance of wild-type *S. pneumoniae* observed at later stages of infection (e.g., Fang et al. 2011, Erttmann et al. 2016, **Fig. 7a-f, Supplementary Fig. 3**).

And lastly, "Does H₂O₂ inhibit bacterial inflammasome activation in vivo, or does it inhibit inflammasome activation by other triggers?" Our analysis indicate that H₂O₂ indiscriminately inhibits inflammasome activation by bacterial components as well as other triggers such as ATP. We have discussed these points in the text (please see lines 97 - 100 and 258 - 263).

Comment #3: Based on Fig 1g-h the others claim that *Streptococcus pneumoniae* infection with the SpxB mutants is cleared by inflammasome activation based on their observation that IL-1 β but not TNF is upregulated in the lungs. However, although only the IL-1 β increase is statistically significant, both IL-1 β and TNF show similar upward trends. I would be a bit more careful with the statement that the stronger innate immune response to the mutant bacteria is inflammasome-dependent (lines 65-67).

Response: We concur with the reviewer. Infection of mice with the H₂O₂-producing *S.p.* WT had an effect on TNF α levels in the lung. However, while IL-1 β levels were significantly increased, TNF α levels were not. IL-1 β is a potent inflammatory molecule that signals *via* the IL-1R leading to the induction of other cytokines. Therefore, changes in IL-1 β secretion could in turn affect NF κ B-mediated cytokines such as TNF α . We have toned down our statement as suggested (please see lines 74 - 76).

Comment #4: In Fig 7i-k it would be good to include the SpxB mutant bacteria in the experiment. The catalase provided intranasally might work also on the host immune system in addition to acting on the bacteria. In order to show that the adverse effects are due to blocking H₂O₂ from the bacteria and not due to effects on the immune system, SpxB mutants should be included.

Response: We thank the reviewer and have now performed the suggested experiment (**Fig. 7l-n**). As predicted, catalase promoted the induction of inflammatory symptoms and the clearance of the H₂O₂-producing *S.p.* WT (**Fig. 7i-k**) but not the *S.p.* Δ *spxB* strain (**Fig. 7l-n**). This is consistent with the conclusion that the observed catalase effect was mainly due to the neutralization of bacteria-derived H₂O₂ rather than to an effect on host immune cells.

Comment #5: The authors put quite some effort in trying to convince the reader that the effects of H₂O₂ lie in the activation step rather than the priming step of inflammasome responses. Judging from the pro-IL1b levels in the western blots I would agree with the authors that the effect is in the activation step (this is also consistent with the effect on ASC oligomerization). However, the authors frequently point to TNF production as a measure for judging inflammasome priming, whereas I think this makes little sense. For instance, the argument for TNF staying the same in Fig 2k-m does not make a lot of sense, since the TNF measured there presumably derives from stimulating the cells with LPS and TLR stimulation by the *S. pneumoniae*. ATP and nigericin themselves do not induce TNF, and *P. aeruginosa* stimulation does not increase the TNF levels that were induced before by the LPS and *S. pneumoniae*. Also the experiments in Sup Fig 2 use TNF secretion as a proxy for inflammasome priming but in the end make little sense. I would invite the authors to think carefully what is needed to address this question. In my opinion, pro-IL-1b levels and the fact that the inhibition occurs already at the level of ASC oligomerization is sufficient proof.

Response: We concur with the reviewer. The pro-IL-1 β levels, the ASC oligomerization data as well as the fact that *S. pneumoniae* blocks the NLRC4 inflammasome, whose activation is independent of priming, are sufficient proof that *S. pneumoniae* affects the activation and not the priming. Nonetheless, to confirm further that the observed effects on inflammasomes are not due to inhibition of priming by *S. pneumoniae*, we pre-treated BMDMs with *S.p.* WT or H₂O₂ before LPS stimulation. Neither *S.p.* WT nor H₂O₂ impeded the expression of pro-IL-1 β or TNF α in such cells (**Supplementary Fig. 2d-f**). In addition, to clarify that priming of inflammasome components is not affected by *S. pneumoniae*, we now also included Western blots for NLRP3 (please see **Fig. 3b,d**, **Fig. 4a**, **Supplementary Fig. 4a** and **Supplementary Fig. 8a**).

Minor comment #1: Streptococci have been reported to activate NLRP3, AIM2 and NLRC4 inflammasomes. Although the authors show an inhibitory effect of *S. pneumoniae* WT upon poly(dA:dT) stimulation, they do not follow up on the AIM2 inflammasome signaling thereafter. This experiment seems to be a bit out of the focus of the story. I would suggest to either omit it, or to further explore whether the inhibitory mechanism is the same for AIM2 as in NLRP3 and NLRC4 inflammasomes?

Response: As recommended by the reviewer, we have now omitted the AIM2 inflammasome data and have instead focussed on NLRP3 and NLRC4 inflammasomes (please see **Fig. 3**).

Minor comment #2: Sup Fig1a: I guess the Y-axis of the IL1b plot should be pg/ml instead of ng/ml.

Response: We apologise and have corrected this error (please see **Supplementary Fig. 1a**).

Minor comment #3: Sup Fig6: Please align the figure labels a bit better.

Response: We have now aligned the labels more thoroughly (please see now **Supplementary Fig. 9**).

Minor comment #4: There are quite some spelling errors in the manuscript, e.g. 'Age- anf sex-matched' on line 253.

Response: We have corrected the spelling errors and thoroughly proofread the manuscript. We thank the reviewer for all the above constructive comments.

Reviewer #2 (Remarks to the Author):

In this study the authors show that while H₂O₂ can facilitate inflammasome assembly, “massive” amounts of H₂O₂ can dampen inflammasome activation. They demonstrate this in the context of infection with *S. pneumoniae*, a pathogen that lacks catalase. The authors show that this phenotype can be reversed by the addition of exogenous catalase both in vitro and in mice. The phenotype is absent in sbx (a pyruvate oxidase) -deficient bacteria correlating with decreased H₂O₂ release in this strain.

This observation is of interest and the experiments presented are overall well-controlled and convincing. I have a few comments to improve the physiological relevance of the study and data interpretation.

Response: We thank the reviewer for the supportive comments and suggestions for improvement. We have addressed all the suggestions as detailed below.

Comment #1: The mechanism of H₂O₂ inhibition is not addressed in the manuscript. The authors suggest in figure S6 that high level of H₂O₂ may oxidize inflammasome components but they do not demonstrate that this may impact activity.

Response: We thank the reviewer for this comment. Detection of Caspase-1 and IL-1 β processing *via* Western blotting is the gold standard for investigating inflammasome activity. Our conclusion is supported by the finding that the inflammasome component ASC is oxidized upon infection with H₂O₂-producing *S. pneumoniae* and that this correlates with reduced ASC oligomerization as well as Caspase-1 and IL-1 β processing. Please note that previous Supplementary Fig 6 is now **Supplementary Fig. 9**.

Comment #2: While the authors show that assembly of inflammasomes is impaired by high doses of H₂O₂, they cannot exclude an effect on signal 1 (priming). As a control and to support the notion that H₂O₂ does not affect the inflammasome competence of the cell it would be important to complete the data in Fig 3 and 4 with expression levels for inflammasome platforms including NLRP3, AIM4 and NLRC4.

Response: We are grateful to the reviewer for this point. To confirm independently that the observed effects on inflammasomes were not due to inhibition of priming by *S. pneumoniae*, we pre-treated BMDMs with *S.p.* WT or H₂O₂ before LPS stimulation. Neither *S.p.* WT nor H₂O₂ impeded the expression of pro-IL-1 β or TNF α (**Supplementary Fig. 2d-f**). As additional controls for priming, we have now also included the suggested blots for NLRP3 (please see **Fig. 3b, d, Fig. 4a, Supplementary Figure 4a and Supplementary Fig. 8a**). We also probed for AIM2 previously in Fig. 3d. However, please note that in line with the suggestion of reviewer #1, the AIM2 data have been omitted from the manuscript but are embedded here in **Fig. II**. As depicted, whereas AIM2 expression was comparable in *S.p.* WT- and *S.p.* Δ *spxB*-infected cells, inflammasome activation was markedly reduced in cells infected with the former. We were not able to probe for NLRC4 because of a lack for a reliable anti-NLRC4 antibody. However, since activation of the NLRC4 inflammasome is independent of priming it is safe to conclude that the effect of *S.p.* WT infection on *P. aeruginosa*-induced NLRC4 activation was not because of priming (e.g., **Fig. 2c, f, i, I, Fig. 3c, e and Fig. 4b, e, f**).

Figure II *S. pneumoniae* blocks the AIM2 inflammasome in a SpxB-dependent manner. Immunoblots of Caspase-1, IL-1 β and AIM2 in cell lysates and supernatants of LPS-primed BMDMs pre-treated with *S.p.* WT or *S.p.* $\Delta spxB$ (MOI 50) and then transfected with 1 μ g/ml poly(dA:dT) for 2 h for AIM2 activation. The results are representative of three independent experiments.

Comment #3: The authors should investigate apoptosis induction in presence of high levels of H₂O₂. Oxidative stress can promote apoptosis and apoptotic caspases can negatively regulate inflammasome activity.

Response: To address this point, we incubated LPS-primed macrophages with increasing concentrations of H₂O₂ followed by activation of the NLRP3 inflammasome by ATP. In addition to Caspase-1 and IL-1 β , H₂O₂ concentrations dose-dependently inhibited Caspase-3 and Caspase-8 cleavage (**Supplementary Fig. 8a**). This observation is consistent with the overall message of the manuscript: It confirms that H₂O₂ is a potent inhibitor not only of inflammatory caspases but also of apoptotic caspases. Caspases are rich in cysteines, hence generally sensitive to oxidative inactivation. Therefore, H₂O₂-mediated inhibition of inflammasomes is not due to induction of apoptosis.

Minor Comment #1: In some of the figure the legend is not clear is. For example in Fig1 the color chart for panel b to e is unclear. The legend is missing for b or does not corresponds to the picture in e.

Response: We thank the reviewer for the comment. We have now clarified the figure legend of **Fig. 1**.

Reviewer #3 (Remarks to the Author):

The modulation of the host immune response by *Streptococcus pneumoniae* largely depends on virulence bacterial factors. Pneumolysin and the capsule are the best studied virulence factors interfering with phagocytosis and modulation of host cells and host response as well. Moreover, pneumococci are producers of hydrogen peroxide (H₂O₂) and the concentrations of H₂O₂ compete out other bacteria in the hot niche. Because H₂O₂ is released in the host environment this chemical compound is also thought to modulate the immune response. Here the authors show that H₂O₂ inhibits inflammasomes by using wild-type and isogenic *spxB* knockout strains. *SpxB* is a pyruvate oxidase and converts pyruvate under aerobic or microaerobic conditions into acetylphosphate, carbon dioxide and H₂O₂. Importantly, pneumococci do not produce a catalase as other Gram-positive or -negative bacteria, however, they are protected against extracellular oxidative stress. This point should be introduced more specifically, because this is a highly important molecular mechanism in the context of H₂O₂ production and resistance against ROS produced by immune cells. Similar to pneumococci other H₂O₂ producing bacteria block inflammasomes dependent on the amount of H₂O₂ released. The authors further validated their data by using *Asc*^{-/-} KO mice. ASC oligomerization is required for activation of the inflammasome complexes. Taken together, the authors unraveled a novel mechanism of deactivation of inflammasomes and immune suppression probably facilitating colonization. Although the study is well done and technically sound, the reviewer has a few important aspects that have to be addressed, probably also experimentally.

Response: We thank the reviewer for these supportive comments and suggestions for improvement.

Specific comments and experimental design

Comment #1: the introduction has been kept extremely focused and short. There are at least two aspects which have to be addressed: 1. how is the pneumococcus protected against H₂O₂ and 2. the amount of H₂O₂ released under in vitro conditions and under in vivo conditions.

Response: We thank the reviewer for the suggestions. We have now discussed these points in the introduction (please see lines 32 - 40) and have mentioned the role played by *SpxB* in both the production of as well as the protection against H₂O₂ (please see lines 61 – 66).

Comment #2: there are other studies (e.g. Huet et al., 2017 Crit Care Med Protective Effect of Inflammasome Activation by Hydrogen Peroxide in a Mouse Model of Septic Shock; Reck Nunes et al 2018 (<https://doi.org/10.1016/j.preghy.2018.07.006>) which show that oxidative stress activates the inflammasome, thereby leading to elevated levels of IL-1 β and TNF- α , enhanced gene expression of NLRP3, caspase 1 or that that increased concentrations of active caspase-1 and interleukin-1 β are related to an increased concentration of hydrogen peroxide. These results should be cross-linked with the results gained in the submitted study.

Response: We thank the reviewer for this point. We have addressed it in the text (please see lines 239 - 248). To recap, indeed many former studies have linked elevated ROS (reactive oxygen species) with inflammatory signalling including inflammasome activation (Cruz et al. 2007, Tschopp and Schroder 2010, Zhou et al. 2010). However, more recent studies show that ROS are required for optimal priming, i.e. the expression of inflammasome components such as NLRP3 and pro-IL-1 β (Bauernfeind et al. 2011, Juliana et al. 2012, Erttmann et al. 2016). In agreement with the present observations, recent reports show that a decrease in ROS production does not result in diminished but in enhanced inflammasome

activation (Meissner et al. 2010, van de Veerdonk et al. 2010, Erttmann et al. 2016). Consistently, excessive generation of ROS and reactive nitrogen species by immune cells have been shown to impede inflammasome activation (Meissner et al. 2008, Hernandez-Cuellar et al. 2012, Mao et al. 2013, Mishra et al. 2013, Erttmann et al. 2016). The observations referred to by the reviewer (Huet et al. 2017, Nunes et al. 2018) can be explained in terms of the role of ROS in inflammasome priming. For example, the reported transcriptional upregulation of IL-1 β and TNF α by H₂O₂ in the study of Nunes *et al.* is consistent with H₂O₂ affecting priming. We have now included these references in the manuscript (please see line 244).

Comment #3: page 5: BMDMs were infected with *S. pneumoniae* post stimulation with LPS and caspase-1 / Il1beta processing were compared. What happens, when BMDMs were not stimulated with LPS?

Response: We thank the reviewer for this comment and have performed the suggested experiment. Due to space limitation, these data are not included in the manuscript but are shown below in **Figure III**. In the absence of LPS priming, we hardly detected any pro-IL-1 β and IL-1 β p17 in BMDMs infected for 12 hours with *S.p.* WT or Δ *spxB*. Caspase-1 processing and secretion were also extremely weak, but more pronounced after infection with *S.p.* Δ *spxB* than *S.p.* WT. Thus, these data confirm that *S. pneumoniae* by itself is a very weak inflammasome activator.

Figure III *S. pneumoniae* elicits weak and delayed inflammasome activation. Immunoblots of Caspase-1, IL-1 β and NLRP3 in cell lysates and supernatants of BMDMs infected with *S.p.* (D39) WT or Δ *spxB* at MOI 50, 100 or 200 for 12 h. One out of two independent experiments is depicted.

Comment #4: Because the strains are positive for pneumolysin: is there any relevance for pneumolysin which is released as well (despite being an intracellular toxin) pneumolysin is known to activate the NLRP3 inflammasome

Response: We thank the reviewer for this comment and have now tested the pneumolysin (PLY) mutant. Our results show that while short (1 h) infections of primed BMDMs with *S.p.* WT or *S.p.* Δ *ply* were unable to induce inflammasome activation, infections for 12 hours led to detectable Caspase-1 and IL-1 β processing by *S.p.* WT but not *S.p.* Δ *ply* (**Supplementary Fig. 3a**). We did not observe any differences between *S.p.* WT and *S.p.* Δ *ply* in their ability to inhibit inflammasome inhibition under co-stimulatory settings (data not shown). As a proxy to study inflammasome activation by PLY, we used listeriolysin O (LLO), a closely related pore-forming toxin. We observed that LLO dose-dependently induced inflammasome activation that was inhibited by *S.p.* WT (**Supplementary Fig. 3b-d**). Thus, although PLY is a potent inflammasome activator, it appears that inflammasome activation by PLY is sensitive to H₂O₂. This is consistent with the restrained inflammasome activation by *S. pneumoniae*.

Comment #5: page 6: *S. pneumoniae* inhibits dose-dependently secretion of caspase-1 etc. Which amounts of H₂O₂ are needed. A kintecs would be nice and in addition, which is the amount or concentration that is needed at least.

Response: We thank the reviewer for this suggestion. We have now included a H₂O₂ dose response on inflammasome inhibition. A significant reduction in Caspase-1 cleavage was observed at 50 to 100 μM of H₂O₂ (please see **Supplementary Fig. 8a**).

Comment #6: the authors have used also a complemented *spxB*-mutant. Are the levels of H₂O₂ production comparable to the wild-type?

Response: We have now included the experimental confirmation that the amounts of H₂O₂ produced by *S.p.* WT and *S.p.* Δ *spxB* + SpxB are comparable and that these H₂O₂ amounts are dramatically reduced in the presence of catalase (please see **Supplementary Fig. 8b, c**).

Comment #7: page 6 - in the experiments represented in Figure 4 the complemented strain should be used as well

Response: We have now included the complemented strain in the assays. Please see **Supplementary Fig. 4**.

Comment #8: the *spxB* knockout showed a growth advantage via the wild-type. I assume this is in BHI, i.e. in a complex medium. How about growth in a minimal medium?

Response: We have now performed the bacterial growth curves in minimal medium (C medium). In minimal medium, *S.p.* Δ *spxB* also has a growth advantage over *S.p.* WT and the complemented *S.p.* Δ *spxB* + SpxB. Catalase increased the growth rate of *S.p.* WT and *S.p.* Δ *spxB* + SpxB but not of *S.p.* Δ *spxB*. This confirms that the observed restrained growth was due to SpxB-generated H₂O₂ (please see below, **Fig. IV**).

Figure IV SpxB-generated H₂O₂ impedes the growth of *S. pneumoniae*. **a** Growth of *S. pneumoniae* D39 WT, Δ *spxB* and Δ *spxB* + SpxB mutants in minimal medium (C medium). **b** Growth curve of *S. pneumoniae* D39 WT, Δ *spxB* and Δ *spxB* + SpxB mutants in minimal medium in the presence or absence of 100 U/ml catalase. Data are representative of two independent experiments performed in technical triplicates. Data are presented as mean \pm SD.

Comment #9: Figure 1. the infection dose should be given in the figure legend;

Response: We have now included the infection dose for each panel in **Fig. 1**.

Comment #10: why is the *spxB*-mutant still producing H₂O₂ (Figure 1)?

Response: *S. pneumoniae* has two enzymes that contribute to H₂O₂ production: pyruvate oxidase and lactate oxidase. Pyruvate oxidase (SpxB) is the main source of H₂O₂ generated by *S. pneumoniae* (Spellerberg et al. 1996, Pericone et al. 2003, Pesakhov et al. 2007, Ramos-Montanez et al. 2008, Echlin et al. 2016). Therefore, *spxB*-deficient strains still show

residual levels of H₂O₂. However, these levels do not seem to be sufficient to inhibit inflammasome activation.

Minor Comment #1: page 11, line 222-223: please check sentence and wording

Response: We have now rephrased the paragraph (please see lines 249 - 253).

Minor Comment #1: page 12, line 253: Age-and-sex...(?: and?)

Response: We have now corrected the literal error (now in line 283).

Minor Comment #1: page 13, line 269: the authors have to describe more precisely in their assays (results) and Figure legends which pneumococcal strain have been used. In addition, the R6 is uncapsulated and can't have any serotype.

Response: We thank the reviewer for these comments. We have now included more detailed culture conditions of the used strains (see lines 298 – 318). Moreover, we indicated thoroughly in all figure legends the *S.p.* strains used.

References (to reviewers #1 - #3)

Bauernfeind, F., et al. (2011). "Cutting edge: reactive oxygen species inhibitors block priming, but not activation, of the NLRP3 inflammasome." J Immunol **187**(2): 613-617.

Cruz, C. M., et al. (2007). "ATP activates a reactive oxygen species-dependent oxidative stress response and secretion of proinflammatory cytokines in macrophages." J Biol Chem **282**(5): 2871-2879.

Echlin, H., et al. (2016). "Pyruvate Oxidase as a Critical Link between Metabolism and Capsule Biosynthesis in *Streptococcus pneumoniae*." PLoS Pathog **12**(10): e1005951.

Erttmann, S. F., et al. (2016). "Loss of the DNA Damage Repair Kinase ATM Impairs Inflammasome-Dependent Anti-Bacterial Innate Immunity." Immunity **45**(1): 106-118.

Fang, R., et al. (2011). "Critical roles of ASC inflammasomes in caspase-1 activation and host innate resistance to *Streptococcus pneumoniae* infection." J Immunol **187**(9): 4890-4899.

Hernandez-Cuellar, E., et al. (2012). "Cutting edge: nitric oxide inhibits the NLRP3 inflammasome." J Immunol **189**(11): 5113-5117.

Huet, O., et al. (2017). "Protective Effect of Inflammasome Activation by Hydrogen Peroxide in a Mouse Model of Septic Shock." Crit Care Med **45**(2): e184-e194.

Juliana, C., et al. (2012). "Non-transcriptional priming and deubiquitination regulate NLRP3 inflammasome activation." J Biol Chem **287**(43): 36617-36622.

Mao, K., et al. (2013). "Nitric oxide suppresses NLRP3 inflammasome activation and protects against LPS-induced septic shock." Cell Res **23**(2): 201-212.

Meissner, F., et al. (2008). "Superoxide dismutase 1 regulates caspase-1 and endotoxic shock." Nat Immunol **9**(8): 866-872.

Meissner, F., et al. (2010). "Inflammasome activation in NADPH oxidase defective mononuclear phagocytes from patients with chronic granulomatous disease." Blood **116**(9): 1570-1573.

Mishra, B. B., et al. (2013). "Nitric oxide controls the immunopathology of tuberculosis by inhibiting NLRP3 inflammasome-dependent processing of IL-1beta." Nat Immunol **14**(1): 52-60.

Nunes, P. R., et al. (2018). "Hydrogen peroxide-mediated oxidative stress induces inflammasome activation in term human placental explants." Pregnancy Hypertens **14**: 29-36.

Pericone, C. D., et al. (2003). "Factors contributing to hydrogen peroxide resistance in *Streptococcus pneumoniae* include pyruvate oxidase (SpxB) and avoidance of the toxic effects of the fenton reaction." J Bacteriol **185**(23): 6815-6825.

Pesakhov, S., et al. (2007). "Effect of hydrogen peroxide production and the Fenton reaction on membrane composition of *Streptococcus pneumoniae*." Biochim Biophys Acta **1768**(3): 590-597.

Ramos-Montanez, S., et al. (2008). "Polymorphism and regulation of the spxB (pyruvate oxidase) virulence factor gene by a CBS-HotDog domain protein (SpxR) in serotype 2 *Streptococcus pneumoniae*." Molecular Microbiology **67**(4): 729-746.

Spellerberg, B., et al. (1996). "Pyruvate oxidase, as a determinant of virulence in *Streptococcus pneumoniae*." Molecular Microbiology **19**(4): 803-813.

Tschopp, J. and K. Schroder (2010). "NLRP3 inflammasome activation: The convergence of multiple signalling pathways on ROS production?" Nat Rev Immunol **10**(3): 210-215.

van de Veerdonk, F. L., et al. (2010). "Reactive oxygen species-independent activation of the IL-1beta inflammasome in cells from patients with chronic granulomatous disease." Proc Natl Acad Sci U S A **107**(7): 3030-3033.

Zhou, R., et al. (2010). "Thioredoxin-interacting protein links oxidative stress to inflammasome activation." Nat Immunol **11**(2): 136-140.

REVIEWERS' COMMENTS:

Reviewer #1 (Remarks to the Author):

The investigators have done a very good job in comprehensively responding to the reviewers' comments and suggestions, including providing additional data that strengthens the conclusions of their study. I recommend publication of this study in Nature Communications, and I only have a couple of minor suggestions for improvement:

1. I personally think that figures I and III from the authors' response letter to the reviewers could be included in the manuscript, since both figures convincingly illustrate that the SpxB mutant bacteria have an intrinsically increased capacity to activate the inflammasome when compared to WT bacteria. This is nicely in line with the novel concept of the paper that bacterial H₂O₂ production limits inflammasome activation.

2. Supplementary Figure 8a contains novel data showing that H₂O₂ affects also LPS/ATP-induced caspase-3 and -8 activation in addition to caspase-1. This is not mentioned or discussed in the manuscript, and it raises some questions about the ASC oxidation that the authors propose as a mechanism by which H₂O₂ inhibits inflammasomes. The fact that also caspase-3 and -8 are affected suggests that oxidation of the catalytic cysteine of caspases could be part of the mechanism by which H₂O₂ inhibits inflammasome activation. Alternatively, it is possible that also caspase-3 and -8 activation upon LPS/ATP stimulation are ASC-dependent. The authors have ASC-deficient mice available, so they could check that. Can the authors please mention and briefly discuss these novel data in the manuscript?

3. Supplementary Figures 3e and 3f are not mentioned or discussed in the manuscript.

4. There is a typo in Figure 2a: nigericin.

5. The authors refer to ASC-deficient mice in the manuscript italicized as *Asc*^{-/-} mice and thus probably referring to the gene name. However, the gene encoding ASC is named *Pycard*. Therefore, officially, when referring to the gene these mice are italicized as *Pycard*^{-/-} mice. However, for these mice it is common to refer to the protein to avoid confusion: non-italicized as ASC^{-/-} mice.

Reviewer #2 (Remarks to the Author):

The revised manuscript is improved. The author addressed many of the comments raised by the reviewers. A weak point remains: what is the mechanism by which H₂O₂ dampen inflammasome activation. In the rebuttal letter, the authors mention that ASC oxidization as a possible mechanism. This refers to Figure supplementary 9c. However, they did little to improve the quality of the pull-down presented in this figure. The absence of MW marker makes difficult to interpret the fuzzy band presented as oxASC. H₂O₂ mostly oxidizes cysteines within cells. ASC has only one cysteine present in the entire protein between the helix 4 and 5 of the CARD. It could have been interesting to improve the significance of the manuscript to interrogate if the mutation of this cysteine abolished ASC oxidation. This could also provide a tool to demonstrate the specificity of the proposed model. As mentioned by the authors, the catalytic site of caspases (including caspase-1) is very sensitive to oxidation as demonstrated before in other studies (<https://www.nature.com/articles/ni.1633.pdf>).

Reviewer #3 (Remarks to the Author):

The authors have nicely and adequately addressed the comments raised by the reviewer. I have no further comments or suggestions that have to be addressed.

Point-by-Point response to the Referees' comments

Reviewer #1 (Remarks to the Author): The investigators have done a very good job in comprehensively responding to the reviewers' comments and suggestions, including providing additional data that strengthens the conclusions of their study. I recommend publication of this study in Nature Communications, and I only have a couple of minor suggestions for improvement:

Response: We thank the reviewer for these supportive comments. We have now included the suggested changes.

Comment 1: I personally think that figures I and III from the authors' response letter to the reviewers could be included in the manuscript, since both figures convincingly illustrate that the SpxB mutant bacteria have an intrinsically increased capacity to activate the inflammasome when compared to WT bacteria. This is nicely in line with the novel concept of the paper that bacterial H₂O₂ production limits inflammasome activation.

Response: We have now included Figures I and III from the authors' response letter in the manuscript. Data are presented in Supplementary Fig. 3. Please see also lines 117 – 119.

Comment 2: Supplementary Figure 8a contains novel data showing that H₂O₂ affects also LPS/ATP-induced caspase-3 and -8 activation in addition to caspase-1. This is not mentioned or discussed in the manuscript, and it raises some questions about the ASC oxidation that the authors propose as a mechanism by which H₂O₂ inhibits inflammasomes. The fact that also caspase-3 and -8 are affected suggests that oxidation of the catalytic cysteine of caspases could be part of the mechanism by which H₂O₂ inhibits inflammasome activation. Alternatively, it is possible that also caspase-3 and -8 activation upon LPS/ATP stimulation are ASC-dependent. The authors have ASC-deficient mice available, so they could check that. Can the authors please mention and briefly discuss these novel data in the manuscript?

Response: We thank the reviewer for this comment. As mentioned by the reviewer, not only Caspase-1 but also Caspase-8 and Caspase-3 processing is affected by stimulation with raising concentrations of H₂O₂. Former studies have demonstrated that caspases can be reversibly inactivated by reactive oxygen species due to their cysteine residues, which are sensitive to oxidation (Borutaite et al. 2001, *FEBS letters*, 500(3), 114-118; Meissner et al. 2008, *Nature Immunology*, 9, 866–872). We therefore assume that H₂O₂ oxidizes not only the single cysteine residue of ASC but also cysteine residues in caspases resulting in inhibition of their catalytic cleavage. We have now described and discussed these data in the manuscript; please see lines 199 – 202.

Comment 3: Supplementary Figures 3e and 3f are not mentioned or discussed in the manuscript.

Response: We have now included the description and conclusion of the above-mentioned figures in the result section. Please note that Supplementary Fig. 3e and 3f are now Supplementary Fig. 4e and 4f, please see lines 133 – 136.

Comment 4: There is a typo in Figure 2a: nigericin.

Response: We thank the reviewer for this note. We have corrected the typo.

Comment 5: The authors refer to ASC-deficient mice in the manuscript italicized as *Asc*^{-/-} mice and thus probably referring to the gene name. However, the gene encoding ASC is named Pycard. Therefore, officially, when referring to the gene these mice are italicized as

Pycard^{-/-} mice. However, for these mice it is common to refer to the protein to avoid confusion: non-italicized as ASC^{-/-} mice.

Response: We thank the reviewer for this comment. We have done the changes accordingly in the main text (lines 207 – 215), Figure 7 and in the legend of Figure 7 (*Asc^{-/-}* to ASC^{-/-}).

Reviewer #2 (Remarks to the Author): The revised manuscript is improved. The author addressed many of the comments raised by the reviewers. A weak point remains: what is the mechanism by which H₂O₂ dampen inflammasome activation. In the rebuttal letter, the authors mention that ASC oxidization as a possible mechanism. This refers to Figure supplementary 9c. However, they did little to improve the quality of the pull-down presented in this figure. The absence of MW marker makes difficult to interpret the fuzzy band presented as oxASC. H₂O₂ mostly oxidizes cysteines within cells. ASC has only one cysteine present in the entire protein between the helix 4 and 5 of the CARD. It could have been interesting to improve the significance of the manuscript to interrogate if the mutation of this cysteine abolished ASC oxidation. This could also provide a tool to demonstrate the specificity of the proposed model. As mentioned by the authors, the catalytic site of caspases (including caspase-1) is very sensitive to oxidation as demonstrated before in other studies (<https://www.nature.com/articles/ni.1633.pdf>).

Response: We thank the reviewer for this comment. To improve the reliability and quality of the oxASC blot, we have now included a less cropped picture of the oxASC blot with indicated MW marker. To clarify, oxidative carbonylation targets not only cysteine but also other amino acids including lysine, arginine, proline, threonine. We have included this point in the text (line 187-189). We have also acknowledged that in addition to ASC, *S. pneumoniae*-mediated inflammasome inhibition is likely involves other components including the oxidation-sensitive cysteine-rich caspases. Please see lines 199 – 202.

Reviewer #3 (Remarks to the Author): The authors have nicely and adequately addressed the comments raised by the reviewer. I have no further comments or suggestions that have to be addressed.

Response: We thank the reviewer for the constructive and supportive comments.